# Optimized Multi-Token Joint Decoding with Auxiliary Model for LLM Inference

**Zongyue Qin**[*]   **Ziniu Hu**[†]   **Zifan He**[*]   **Neha Prakriya**[*]   **Jason Cong**[*]   **Yizhou Sun**[*]

## Abstract

Large language models (LLMs) have achieved remarkable success across diverse tasks, but, due to single-token generation at each decoding step, their inference processes are hindered by substantial time and energy demands. While previous methods such as speculative decoding mitigate these inefficiencies by producing multiple tokens per step, each token is still generated by its single-token distribution. Although this enhances the speed, it does not improve the output quality. In contrast, our work simultaneously boosts inference speed and improves the output effectiveness. We consider multi-token joint decoding (MTJD), which generates multiple tokens from their joint distribution at each iteration, theoretically reducing perplexity and raising task performance. However, MTJD suffers from the high cost of sampling from the joint distribution of multiple tokens. Inspired by speculative decoding, we introduce multi-token assisted decoding (MTAD), a novel framework designed to accelerate MTJD. MTAD leverages a smaller auxiliary model to approximate the joint distribution of a larger model, incorporating a verification mechanism that not only ensures the accuracy of this approximation, but also increases the decoding efficiency over conventional speculative decoding. Theoretically, we demonstrate that MTAD closely approximates exact MTJD with a bounded error. Empirical evaluations across various tasks reveal that MTAD improves downstream performance by 25% compared to standard single-token sampling. Furthermore, MTAD achieves a $1.42\times$ speed-up and consumes $1.54\times$ less energy than vanilla speculative decoding methods. These results highlight MTAD's ability to make multi-token joint decoding both effective and efficient, promoting more productive and high-performance deployment of LLMs.[1]

## 1  Introduction

Large Language Models (LLMs) such as GPT-4 and Llama-2 (Touvron et al., 2023) have demonstrated extraordinary capabilities across a wide range of tasks (Brown et al., 2020; Chowdhery et al., 2023; Thoppilan et al., 2022; Touvron et al., 2023). Despite their impressive performance, the deployment of LLMs is often constrained by substantial inference costs in terms of time and energy. This inefficiency primarily stems from the autoregressive nature of these models, where generating a sequence of $K$ tokens requires $K$ separate model calls. Each call involves loading large weight matrices and intermediate results from GPU global memory to computing units, leading to repeated memory accesses and limited hardware utilization (Samsi et al., 2023; Leviathan et al., 2023).

To tackle this challenge, researchers have delved into non-autoregressive decoding approaches. Early methods (Ghazvininejad et al., 2019; Gu et al., 2017; Guo et al., 2020) aimed at reducing inference latency by concurrently generating multiple tokens. But these methods usually require task-dependent techniques and information to match the performance of autoregressive decoding (Kim et al., 2023; Xiao et al., 2023). More recently, speculative decoding has emerged (Leviathan et al., 2023; Chen et al., 2023; Kim et al., 2023; Sun et al., 2023). This method exploits the observation that most of the small model's prediction aligns well with that of a large model. It leverages a smaller auxiliary model to draft a few future tokens autoregressively, which are subsequently validated in parallel by the

---

[*]Department of Computer Science, University of California, Los Angeles, USA. Correspondence to: qinzongyue@cs.ucla.edu

[†]California Institute of Technology, USA.

[1]We release our code at https://github.com/ZongyueQin/MTAD

larger model. As the smaller model operates significantly faster and parallel token verification incurs a similar time cost as generating a single token, speculative decoding attains an overall speed-up of 1-2×. Despite gains in speed, these methods still generate each token based on its single-token probability. Consequently, it does not enhance the effectiveness of the generated sequences.

In this work, we first go beyond the conventional trade-off between efficiency and effectiveness and explore multi-token joint decoding (MTJD). Unlike traditional approaches, MTJD produces multiple tokens from their joint distribution at each decoding step. Theoretically, we show this joint generation can lead to lower perplexity and hence improved task performance. However, directly sampling from the joint distribution of multiple tokens poses significant computational challenges, rendering MTJD impractical.

Inspired by speculative decoding, we propose multi-token assisted decoding (MTAD), a novel framework designed to approximate and accelerate MTJD. MTAD employs a smaller auxiliary model to estimate the joint distribution of a larger model, significantly reducing computational demands. To ensure the accuracy of this approximation, MTAD incorporates a verification mechanism that not only guarantees the accuracy of the draft tokens but also enhances efficiency beyond conventional speculative decoding by maximizing the number of accepted tokens per iteration. We provide both theoretical and empirical analyses to demonstrate that MTAD boosts perplexity and downstream performance. Meanwhile, it significantly reduces the energy and time usage compared to existing decoding strategies.

Our contributions are as follows:

1. We introduce multi-token joint decoding (MTJD), a multi-token joint decoding approach that theoretically reduces perplexity by generating tokens from their joint distribution.

2. We develop multi-token assisted decoding (MTAD), an efficient approximation of MTJD with bounded error that leverages a smaller model for distribution approximation.

3. We analyze the energy consumption of LLM inference. To our knowledge, we are the first to give quantified and empirical evidence that, despite the fact that MTAD and other speculative decoding algorithms increase the number of FLOPs needed during LLM inference, they actually use less energy with fewer accesses to the GPU global memory.

4. We conducted comprehensive evaluations across various tasks, demonstrating that MTAD improves downstream performance by 25% compared to standard single-token sampling, while, at the same time, realizing a 1.42× speed-up and paring energy consumption by 1.54× compared to vanilla speculative decoding methods.

These advancements position MTAD as a robust solution for making multi-token joint decoding both effective and efficient, thereby facilitating more sustainable and high-performance deployment of large-scale language models.

## 2 PRELIMINARIES

In this section, we discuss preliminaries relevant to contextualizing our paper.

### 2.1 DECODINGS OF LLMS

**Decoding and Perplexity.** Let $p$ denote the distribution defined by LLM model $M_p$. Given an input context $input$, a decoding algorithm generates a sequence of $N$ tokens whose likelihood is designated as $p(x_{1:N}|input)$. The likelihood of the sequence is directly linked to the *perplexity* of the sequence, which is the exponentiated average negative log-likelihood of all tokens. Based on autoregressive decomposition $p(x_{1:N}|input) = \prod_{t=1}^{N} p(x_t|x_{1:t-1}, input)^2$, the perplexity is defined as:

$$PPL(x_{1:N}) = \exp\left\{-\frac{1}{N}\sum_{t=1}^{N}\log p(x_t|x_{1:t-1})\right\} \quad (1)$$

---

[2]In the paper, we omit $input$ when there is no ambiguity.

Perplexity serves as a direct metric for assessing the effectiveness of a decoding algorithm. In practice, when a model is well-trained, lower perplexity often correlates with improved downstream performance. For example, beam sampling (explained below) aims to return output with lower perplexity and is empirically proven to have better downstream performance in general (Shi et al., 2024).

To further demonstrate the relationship between perplexity and downstream performance, we evaluate GPT-3.5-turbo on the spider (Yu et al., 2018) dataset. Employing a temperature of 2, the model generated 10 outputs for each input. We measured the average perplexities and execution accuracies for the outputs with the highest, lowest, and median (the 5th lowest) perplexity. As seen in Table 1, lower perplexity correlates with improved downstream performance, even in one of today's largest models.

Now we introduce commonly used decoding approaches.

**Multinomial Sampling.** Multinomial sampling, also known as standarized sampling or single-token sampling, samples the next token $x_t$ based on $\mathcal{T} \circ p(\cdot|x_{1:t-1}, input)$, where $\mathcal{T}$ is a warping operation applied to enhance the high probability region. Some common warping operations include *top-k* warping. This limits the selection to the top k tokens, and *top-p* warping, where tokens are sampled from the smallest possible subset of the vocabulary whose cumulative probability mass exceeds a specified threshold.

Table 1: Relationship between perplexity and execution accuracy (EA, higher the better) for GPT-3.5-turbo.

| Output | Avg. PPL ↓ | EA (%) ↑ |
|---|---|---|
| Highest PPL | 4.13 | 33 |
| 5-th Lowest PPL | 1.40 | 58 |
| Lowest PPL | 1.07 | 62 |

The deterministic version of multinomial sampling is a special case with $k = 1$, also called greedy decoding.

**Beam Sampling.** Beam sampling is intended to decrease output perplexity over multinomial sampling. For each position $t$ ($1 \leq t \leq N$), it maintains $W > 1$ candidate sequences, which are also called *beams*. Assume we have already kept the $W$ sequences $\mathcal{I}_{t-1} = \{x_{1:t-1}^{(1)}, \ldots, x_{1:t-1}^{(W)}\}$ at position $t - 1$. $W$ sequences with length $t$ are then sampled from $\mathcal{T} \circ p_{beam}$, where $p_{beam} : \mathcal{I}_{t-1} \times V \to [0, 1]$ is the beam sampling probability:

$$p_{beam}(x_{1:t-1}^{(i)}, x_t) = \frac{p(x_{1:t-1}^{(i)}, x_t|input)}{\sum_{1 \leq j \leq W, x_t' \in V} p(x_{1:t-1}^{(j)}, x_t'|input)} \tag{2}$$

Notice that $p(x_{1:t-1}^{(i)}, x_t|input) = p(x_t|x_{1:t-1}^{(i)}, input) \cdot p(x_{1:t-1}^{(i)}|input)$. In practice, beam sampling stores the likelihood $p(x_{1:t-1}^{(i)}|input)$ for each beam, and the computation complexity of $p_{beam}$ is $O(W \cdot |V|)$. In deterministic beam sampling, the top $W$ sequences with the highest likelihood $p_{beam}(x_{1:t})$ will be kept.

## 2.2 VANILLA SPECULATIVE DECODING

Besides effectiveness, speculative decoding is proposed by (Leviathan et al., 2023; Chen et al., 2023) to accelerate the inference of LLMs. It utilizes a small model to generate the next $\gamma$ tokens and then uses the large model to verify the drafted tokens *in parallel*, which is summarized below:

1. Let *input* be the input context, the small model samples $\gamma$ draft tokens $x_1, \ldots, x_\gamma$ with multinomial sampling based on $\tilde{q}(x_t|x_{1:t-1}, input))$ for $t = 1, \ldots, \gamma$, where $\tilde{q} = \mathcal{T} \circ q$ and $q$ is the small model's output distribution.

2. The large model verifies the draft tokens in parallel by computing the conditional probability $\tilde{p}(x_t|x_{1:t-1}, input)$ for $t = 1, \ldots, \gamma$.

3. Each draft token $x_t$ is accepted with probability $\min(1, \tilde{p}(x_t)/\tilde{q}(x_t))$. The draft tokens before the first rejected token are kept as the decoding output. An additional token is sampled from a residual distribution as a correction to the first rejected token. Then the accepted tokens and the resampled token are appended to the context *input* as the input to the next iteration.

4. Repeat step 1-3 until reaching the stopping criteria, e.g., reaching the length limit.

Because the large model verifies $\gamma$ tokens in parallel with one run, the time cost is smaller than calling it $\gamma$ times. Moreover, the global memory access is also pared, which saves energy consumption, as we shall illustrate in Section 4. Meanwhile, although the small model still runs in an autoregressive way, its inference speed is more efficient than the large model. As a result, speculative decoding maintains an identical sampling distribution while realizing a speedup of 1–2$\times$ compared to multinomial sampling and using less energy.

## 3 METHODOLOGY

As discussed in Section 2, the goal of this work is to design an algorithm that yields lower perplexity and better efficiency than multinomial sampling and vanilla speculative decoding. In this section, we first introduce multi-token joint decoding (MTJD). This generates multiple tokens based on their joint likelihood. We prove it can yield lower perplexity. Then we present multi-token assisted decoding (MTAD), which approximates and accelerates MTJD by exploiting an auxiliary model.

### 3.1 MULTI-TOKEN JOINT DECODING

We first explain a new decoding algorithm to improve multinomial sampling in terms of perplexity.

**Definition 3.1. Multi-Token Joint Decoding**. Let $M_p$ be the large target model with distribution $p$. Different from single-token multinomial sampling, multi-token joint decoding (MTJD) produces the next $\gamma_i$ tokens at step $i$ based on their joint conditional probability $p(x_{t+1:t+\gamma_i}|x_{1:t})$, where $\gamma_i$ is an integer no less than 1 and $t = \sum_{i'=1}^{i-1} \gamma_{i'}$, i.e., the total tokens generated in the previous $i-1$ steps.

Multinomial sampling is a special case of MTJD where $\gamma_i = 1$, $\forall i$. When $\gamma_1 = N$, MTJD generates the sequence directly based on their joint likelihood. So intuitively, output perplexity should improve as $\gamma_i$ increases. Besides, generating $\gamma_i$ tokens simultaneously allows MTJD to consider their interactions. In contrast, multinomial sampling selects each token without considering any future tokens. So MTJD is less prone to choosing local optima.

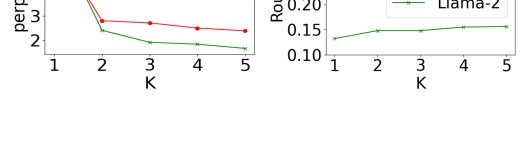

Figure 1: Perplexity and Rouge-L score of the output when $\gamma_i = K$ for MTJD with OPT-125M and Llama-2-68M fine-tuned on ChatGPT-Prompts (Rashad, 2023) dataset.

Theorem 3.2 demonstrates the limit of perplexity of MTJD when $N$ approaches infinity. The proofs are included in the Appendix A.

**Theorem 3.2.** *Assume at the $i$-th $(i = 1, \ldots, N)$ iteration, MTJD generates $\gamma_i$ tokens. Let $\Gamma_i$ denote the total number of tokens generated at the first $i$ iterations. Let $x_{1:\Gamma_N}$ denote the generated tokens. When $N \to \infty$*

$$PPL_p(x_{1:\Gamma_N}) \to \exp\left(-\frac{1}{\bar{\gamma}}\mathbb{E}_\gamma L_p(\gamma, \tilde{p})\right) \tag{3}$$

*where $\bar{\gamma}$ is the expected number of $\gamma_i$, $\tilde{p} = \mathcal{T} \circ p$ represents how we sample the next $\gamma_i$ tokens from $p$ (e.g., in deterministic sampling, $\tilde{p} = \arg\max \circ p$ always returns the tokens with the highest joint likelihood), and $L_p(\gamma, \tilde{p})$ is the expected log-likelihood of the $\gamma$ tokens sampled from $\tilde{p}$:*

$$L_p(\gamma, \tilde{p}) = \mathbb{E}_{x_{1:t} \in \mathcal{X}} \sum_{x_{t+1:t+\gamma}} \tilde{p}(x_{t+1:t+\gamma}|x_{1:t}) \log p(x_{t+1:t+\gamma}|x_{1:t}) \tag{4}$$

*Here $\mathcal{X}$ is the space of all possible inputs.*

**Corollary 3.3.** *Based on Theorem 3.2, we can show that when $N \to \infty$, greedy MTJD (i.e., top-1 MTJD sampling) has lower perplexity than greedy decoding (top-1 single-token sampling).*

Empirical evidence supports our claim. We fine-tune both a Llama and an OPT model on the ChatGPT-Prompts dataset and evaluate the output perplexity and Rouge-L scores with example outputs. Figure

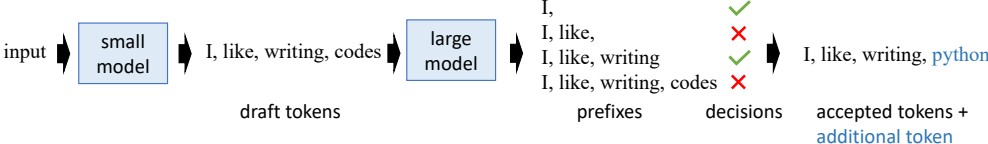

Figure 2: An example of MTAD's verification process. MTAD accepts the *longest* draft sub-sequence that passes verification based on joint likelihood.

1 depicts the output perplexity and Rouge-L scores of MTJD with $\gamma_i$ set to a constant $K$, where $K = 1, \ldots, 5$. Notice that setting $K = 1$ is equivalent to multinomial sampling. We use beam sampling to approximate the $\arg\max$ sampling from the joint distribution $p(x_{t+1:t+K}|x+1:t, input)$. We can see that the perplexity keeps dropping when $K$ increases. It confirms our claim that increasing $\gamma_i$ will increase the output perplexity. Moreover, the Rouge-L score also improves with $K$, supporting our claim that better perplexity reflects enhanced performance in downstream tasks.

### 3.2 MULTI-TOKEN ASSISTED DECODING

Unfortunately, the computation cost of MTJD is infeasible in practice, since the time and space complexity to compute the joint distribution of $\gamma_i$ tokens is $|V|^{\gamma_i}$. Inspired by speculative decoding and the fact that "even when a small model is an order of magnitude smaller than a large model, only a small fraction of the small model's prediction deviate from those of the large model" (Leviathan et al., 2023; Kim et al., 2023), we propose multi-token assisted decoding (MTAD), which exploits a small auxiliary model $M_q$ to accelerate MTJD approximately. The core idea is to (1) use the joint distribution $q(x_{t+1:t+\gamma_i}|x_{1:t})$ output by $M_q$ to approximate $p(x_{t+1:t+\gamma_i}|x_{1:t})^3$ and produce $\gamma$ draft tokens from $q(x_{t+1:t+\gamma_i}|x_{1:t})$, then (2) utilize the large model to validate draft tokens in parallel and accept the *longest* draft prefix sub-sequence that passes verification, and (3) sample an additional token from the distribution of the large model without extra overhead to ensure at least one token is generated at each iteration. However, it is still infeasible to directly generate draft tokens from the joint distribution $q(x_{t+1:t+\gamma_i}|x_{1:t})$. So we propose to further approximate this process with beam sampling, which is an effective and efficient algorithm to generate sequences with high likelihood. In this way, MTAD decreases the number of runs of the large model to generate $N$ tokens, thus accelerating the inference in the same way as vanilla speculative decoding does. Algorithm 1 in the Appendix illustrates the pseudocode of MTAD algorithm.

**Draft Tokens Verification** Figure 2 displays the verification process of MTAD. Let $x_{t+1}, \ldots, x_{t+\gamma}$ be the draft tokens generated by beam sampling with the auxiliary model. Since beam sampling is a widely recognized algorithm to produce sequences with high overall likelihood (Leblond et al., 2021), it is reasonable to assume $q(x_{t+1:t+\gamma}|x_{1:t})$ is large. Also, since beam sampling works in an autoregressive way, we can also infer that $\forall j \in \{1, \ldots, \gamma\}$, $q(x_{t+1:t+j}|x_{1:t})$ is large. To approximate MTJD, for each step $i$, MTAD needs to ensure the accepted tokens $x_{t+1:t+\gamma_i}$ ($0 \le \gamma_i \le \gamma$) also have high joint likelihood with the large model $M_p$. So MTAD first computes the joint likelihood $p(x_{t+1:t+j}|x_{1:t})$ for $j = 1, \ldots, \gamma$. Then for each prefix sub-sequence $x_{t+1:t+j}$, it passes verification if and only if $\min(1, \frac{p(x_{t+1:t+j}|x_{1:t})}{q(x_{t+1:t+j}|x_{1:t})}) > \tau$, where $\tau \in [0, 1)$ is a pre-defined threshold. Notice that if $\min(1, \frac{p(x_{t+1:t+j}|x_{1:t})}{q(x_{t+1:t+j}|x_{1:t})}) > \tau$, we have $\frac{p(x_{t+1:t+j}|x_{1:t})}{q(x_{t+1:t+j}|x_{1:t})} > \tau$, which means $\frac{q(x_{t+1:t+j}|x_{1:t}) - p(x_{t+1:t+j}|x_{1:t})}{p(x_{t+1:t+j}|x_{1:t})} < \frac{1}{\tau} - 1$. Therefore, our acceptance policy guarantees that when $q(x_{t+1:t+j}|x_{1:t}) > p(x_{t+1:t+j}|x_{1:t})$, the relative error is bounded. And if $q(x_{t+1:t+j}|x_{1:t}) \le p(x_{t+1:t+j}|x_{1:t})$, it means the sub-sequence has higher likelihood in the large model, then it is reasonable to accept it. After verifying all the sub-sequences, MTAD accepts the *longest* prefix sub-sequence that passes verification.

The verification step of MTAD ensures that the accepted tokens have a high joint likelihood with the large model. We have shown that selecting multiple tokens based on their joint likelihood leads

---

[3]It is also valid to approximate $\tilde{p}$ with $\tilde{q}$. Without loss of generality, we consider non-warped distribution in the illustration of MTAD.

to better output perplexity. Thus, MTAD is more effective than multinomial sampling and vanilla speculative decoding. Furthermore, since MTAD accepts the longest draft sub-sequence with high likelihood, it can tolerate low-quality tokens as long as the joint likelihood is high. So at each iteration, MTAD can admit more draft tokens than vanilla speculative decoding, which results in better efficiency.

Next, we theoretically analyze the approximation error of MTAD. Lemma 3.4 shows the upper bound of MTAD's perplexity. Theorem 3.5 reveals the upper bound of the ratio between the perplexity of approximate MTAD and exact MTJD. The proofs are given in Appendix A.

**Lemma 3.4.** *Let us assume that when the small auxiliary model generates draft tokens with beam sampling, the beam width is large enough such that the returned log-likelihood is close to the maximum log-likelihood, i.e.,*

$$\mathbb{E}_{x_{1:\Gamma_{i-1}} \in \mathcal{X}} \log q(x_{\Gamma_{i-1}+1:\Gamma_i-1}|x_{1:\Gamma_{i-1}}) \geq$$
$$(1 - \epsilon)\mathbb{E}_{x_{1:\Gamma_{i-1}} \in \mathcal{X}} \left( \max_{x_{\Gamma_{i-1}+1:\Gamma_i-1}} \log q(x_{\Gamma_{i-1}+1:\Gamma_i-1}|x_{1:\Gamma_{i-1}}) \right) \tag{5}$$

*where $\epsilon$ is an error term and $\epsilon \leq 0$ because $\log q \leq 0$.*

*Furthermore, let $H(p,q)$ the single-token cross entropy between $p$ and $q$, i.e., $H(p,q) = -\mathbb{E}_{x_{1:t} \in \mathcal{X}} \sum_{x_{t+1}} p(x_{t+1}|x_{1:t}) \log q(x_{t+1}|x_{1:t})$.*

*With the two assumptions above, when $N \to \infty$ we have*

$$PPL_q(x_{1:\Gamma_N}) \leq \exp(-\frac{1-\epsilon}{\bar{\gamma}}\mathbb{E}_\gamma L_q(\gamma - 1, \arg\max \circ q) + \frac{H(p,q)}{\bar{\gamma}}) \tag{6}$$

*where*

$$L_q(\gamma, \arg\max \circ q) = \mathbb{E}_{x_{1:t} \in \mathcal{X}} \max_{x_{t+1:t+\gamma}} \log q(x_{t+1:t+\gamma}|x_{1:t})) \tag{7}$$

**Theorem 3.5.** *Let $x_{1:\Gamma_N}$ be the tokens generated by approximate MTAD, and $x^*_{1:\Gamma_N}$ be the tokens generated by deterministic exact MTJD. Assume $\forall x_{1:t} \in \mathcal{X}$, $\|\log p(x|x_{1:t}) - \log q(x|x_{1:t})\|_\infty \leq U$, where $U$ is a constant. We have*

$$\lim_{N \to \infty} \frac{PPL_p(x_{1:\Gamma_N})}{PPL_p(x^*_{1:\Gamma_N})} \leq \tau^{-\frac{1}{\bar{\gamma}}} \exp\left( \frac{(1 - \epsilon\bar{\gamma})H(p) + (1 - \epsilon + \bar{\gamma})U}{\bar{\gamma}} \right) \tag{8}$$

*where $H(p)$ is the entropy of $p$ and $\epsilon < 0$ is the error term of beam sampling (see Lemma 3.4).*

Theorem 3.5 suggests the approximation error of MTAD is bounded by a factor related to the verification threshold $\tau$, average number of accepted tokens $\bar{\gamma}$, the difference between the large and small models (measured by $U$), the error of beam sampling $\epsilon$, and the entropy of the large model itself. In addition, the following theorem analyzes $\bar{\gamma}$. The proof is illustrated in Appendix A.

**Theorem 3.6.** *Following the assumption in Theorem 3.5, we have $\bar{\gamma} \geq \frac{|\log \tau|}{U}$.*

With Theorem 3.6, we observe that when $q \to p$, we have $U \to 0$ and $\bar{\gamma} \to \infty$. Meanwhile, when the beam width for the auxiliary model is large enough, $\epsilon \to 0$, and the ratio bound in Theorem 3.5 converges to 1, This implies that MTAD converges to MTJD under these limiting conditions.

Similar to Spectr (Sun et al., 2023) and SpecInfer (Miao et al., 2023), it is possible to enhance the number of accepted tokens in MTAD by allowing the draft model to produce multiple draft sequences and by applying tree-based attention (Miao et al., 2023) for simultaneous verification. However, our preliminary experiment results suggest that since MTAD already selects the longest accepted prefix sub-sequence, the advantage of generating multiple draft tokens is less significant. Moreover, this approach increases the memory cost during inference and may affect the error bounds derived above. Therefore, we leave a more detailed exploration of this extension as future work.

## 4 ENERGY EFFICIENCY ANALYSIS

Previous studies (Leviathan et al., 2023; Chen et al., 2023; Kim et al., 2023; Sun et al., 2023) only focus on the speed of speculative decoding. However, an equally important consideration is energy

Table 2: The effect of batch size to inference speed and energy consumption. The number of inputs is the product of the number of LLM runs and input batch size.

| Batch Size | Energy (J) | Energy/run (J) | Energy/Input (J) | Time (s) | Time/run (s) | Time/input (s) |
|---|---|---|---|---|---|---|
| 1 | 42,450 | 14.1 | 14.1 | 1,129 | 0.376 | 0.376 |
| 2 | 49,621 | 16.5 | 8.26 | 1,191 | 0.397 | 0.198 |
| 4 | 53,325 | 17.7 | 4.43 | 1,178 | 0.392 | 0.098 |
| 8 | 59,210 | 19.7 | 2.46 | 1,211 | 0.403 | 0.050 |
| 16 | 74,058 | 24.7 | 1.54 | 1,255 | 0.418 | 0.026 |

consumption. To our knowledge, there is no existing work evaluating the impact of speculative decoding on inference energy consumption. Although MTAD and speculative decoding raise the number of FLOPs due to the involvement of a small auxiliary model and the rollback operation, they concurrently reduce the inference time and memory operations, which are key factors of GPU (or TPU) energy consumption (Allen & Ge, 2016; Chen et al., 2011). Consequently, it poses an open question regarding whether speculative decoding increases or decreases overall energy consumption.

To understand the net effect of speculative decoding, we decompose the total energy consumption into two parts following (Allen & Ge, 2016):

$$E_{total} = PW_{flop}T_{flop} + PW_{mem}T_{mem} \qquad (9)$$

where $PW_{flop}, PW_{mem}$ denote the power (energy/second) of a unit FLOP and memory operation, respectively, and $T_{flop}, T_{mem}$ are the total time spent on these operations. When input batch size increases, $PW_{flop}$ rises until it reaches the power of maximum FLOPs, designated as $PW_{flop}^*$. $PW_{mem}$ is irrelevant to the input batch size because it only depends on the memory hardware.

To determine the relative magnitude relationship between $PW_{flop}$ and $PW_{mem}$, we first point out the fact that GPU memory operations in LLM inference are dominated by accessing off-chip global memory. This consumes about $100\times$ of energy compared to accessing on-chip shared memory (Jouppi et al., 2021). Because each multiprocessor on a GPU usually has 64KB of on-chip memory shared by multiple threads, but to store a single layer of LLM, say T5-11b (Raffel et al., 2020), requires about 1GB of memory. Moreover, Allen and Ge showed that doing a sequential read from off-chip memory consumes 20-30% more power than running maximum FLOPs (Allen & Ge, 2016). So we have $PW_{mem} > PW_{flop}^* \geq PW_{flop}$. No-

Table 3: Speed and energy cost of multinomial sampling (ms) and speculative decoding (spec).

| | OPT | | LLAMA-2 | |
|---|---|---|---|---|
| | MS | SPEC | MS | SPEC |
| TOKENS/S | 23.8 | 35.6 | 22.0 | 31.6 |
| J/TOKEN | 11.3 | 5.74 | 11.2 | 6.97 |

tice that $PW_{flop}^* = PW_{flop}$ only if the batch size reaches the maximum parallelization capacity of GPUs. During multinomial sampling and speculative decoding, the batch size is generally small (Leviathan et al., 2023). So most of the computing power is not utilized (Leviathan et al., 2023), which means $PW_{mem} \gg PW_{flop}$.

In addition, previous studies have revealed that during LLM inference $T_{mem} \gg T_{flop}$ (Leviathan et al., 2023). Therefore, the energy induced by memory operations, i.e., $PW_{mem}T_{mem}$ dominates $E_{total}$. Since speculative decoding lowers $T_{mem}$ by reducing the number of runs of the large model, it should cut the inference energy consumption to a similar extent as it reduces time consumption.

To validate our hypothesis, we conducted an experiment to evaluate how batch size influences energy consumption during inference. We ran OPT-13b models on a Nvidia L40 GPUs with 48GB memory. Fixing the total number of runs of the large model while varying the input batch size $b \in \{1, 2, 4, 8, 16\}$ for each run, we measured time and energy cost. The details of energy measurement are illustrated in the Appendix D. Table 2 shows the results. As batch size doubles, although the number of FLOPs doubles, the energy consumption per run goes up slightly. This observation demonstrates that $PW_{mem}T_{mem}$ dominates $E_{total}$. Moreover, we measured the speed and energy consumption of running multinomial sampling with the large model and speculative decoding with OPT (125M, 13B) and Llama-2 (68M, 13B) models. The results, seen in Table 3, indicate that speculative decoding lowers the energy consumption and the time cost. This observation corroborates our claim to the energy efficiency of speculative decoding.

## 5 EXPERIMENTS

**Datasets and Models**. In the main paper, we report results with three public datasets for evaluation: (1) Spider (Yu et al., 2018), MTBench (Zheng et al., 2023), and HumanEval (Chen et al., 2021). We use Llama-3-8B and Llama-3-8B-Instruct (Dubey et al., 2024) as target models, and Llama-3-1B and Llama-3-1B-Instruct as their draft models, respectively. We provide additional experiments with other datasets and model families in Appendix C.

**Baselines**. We compare our method with six speculative decoding methods, including four lossless decoding methods: vanilla speculative decoding (*SpD*) (Lee et al., 2018; Chen et al., 2023), *Spectr* (Sun et al., 2023), *SpecInfer* (Miao et al., 2023), *MCSS* (Yang et al., 2024), and two lossy speculative decoding methods: *BiLD* (Kim et al., 2023) and *typical decoding* (Cai et al., 2024). All the baselines and our method utilize the same pair of draft and target models without any fine-tuning. For each method, we let it generate at most 128 tokens for each input and run it for $1,000$ seconds. All the methods are stochastic with top-$k$ and top-$p$ sampling with the temperature $= 1$. The details of the hyper-parameters (e.g., $k$ and $p$) and machine configurations of the experiments can listed in the Appendix D, E, and F.

Appendix C reports additional experiments and ablation studies.

### 5.1 PERFORMANCE OF MULTI-TOKEN JOINT DECODING

While most speculative decoding approaches focus on inference speed up, we want to design approaches that can also improve inference quality. We propose multi-token joint decoding (MTJD, Section 3.1) to accomplish the goal, due to its capability to achieve a lower perplexity and higher likelihood than single-token multinomial sampling. To validate that MTJD indeed betters output quality, we test MTJD (k=4) and standard multinomial sampling on Spider, MTBench, and HumanEval using the Llama-3 series models. We follow the same way introduced in Section 3.1 to implement MTJD. For this process, the higher the scores, the better the downstream performance. Under all settings, MTJD realizes the highest scores and lower perplexity. These results show a clear advantage for MTJD in terms of output quality.

Table 4: Performance comparison of single-token sampling and multi-token joint sampling. We use Llama-3.1-8B and Llama-3.1-8B-Instruct as target models, and Llama-3.2-1B and Llama-3.2-1B-Instruct as the draft models.

|  | Llama-3 (8B,1B) | | | Llama-3-Instruct (8B,1B) | | |
|---|---|---|---|---|---|---|
|  | Spider | MTBench | HumanEval | Spider | MTBench | HumanEval |
| *Single-token multinomial sampling* | | | | | | |
| *Score* | 22.0 | 3.40 | 15.9 | 36.0 | 4.11 | 28.0 |
| *PPL* | 2.58 | 2.40 | 2.09 | 2.23 | 1.91 | 1.85 |
| *Multi-token joint sampling* | | | | | | |
| *Score* | 52.5 | 3.77 | 36.6 | 60.5 | 4.40 | 49.4 |
| *PPL* | 1.16 | 1.32 | 1.26 | 1.18 | 1.27 | 1.15 |

### 5.2 PERFORMANCE OF MULTI-TOKEN ASSISTED DECODING

Next, we evaluate the efficiency and effectiveness of MTAD, an approximate algorithm that accelerates MTJD while preserving its downstream performance advantages. Table 5 presents the decoding speed, energy consumption, and downstream performance of various decoding algorithms across different datasets.

**Efficiency Analysis.** We first observe that MTAD is the most efficient among all baselines in terms of both energy and time. On average, MTAD is **17.2% faster** than the most efficient lossless baseline, MCSS, while consuming **6.1%** less energy. Compared to lossy decoding algorithms, it attains **27.2% (15.1%)** higher speed and **17.9% (14.1%)** lower energy consumption than BiLD (typical decoding), respectively. Interestingly, despite utilizing only a single draft sequence, MTAD

outperforms baselines that employ multiple draft sequences, such as Spectr, MCSS, and SpecInfer. In these methods, verification terminates immediately upon rejecting a token. In contrast, MTAD continues verification even after a rejection, searching for future tokens that may still pass. This mechanism results in a greater acceptance length per iteration than the baselines.

**Downstream Performance Comparison.** Next, we compare the downstream performance of different decoding algorithms. Notice that while lossless decoding algorithms theoretically sample from the target distribution, they exhibit slight variations in downstream performance. This discrepancy arises because, despite preserving the original distribution, differences in the sampling process prevent them from generating identical sequences even when the random seed is fixed. Furthermore, we observe that lossy decoding algorithms can reach higher downstream performance at the expense of efficiency. This suggests that all lossy decoding methods can trade off efficiency for performance by adjusting verification strictness. Most notably, MTAD consistently achieves the highest downstream performance. On average, it surpasses lossless decoding algorithms by **25.0%**, BiLD by **11.8%**, and typical decoding by **9.9%**. These results confirm our claim that MTAD offers superior effectiveness compared to conventional decoding methods that rely solely on single-token distributions.

Table 5: Comparison of different speculative decoding methods across various models and metrics. Bold indicates best values, underline indicates second-best.

| | Lossy Decoding | | Lossless Decoding | | | | Ours |
|---|---|---|---|---|---|---|---|
| | BiLD | Typical | SpD | Spectr | SpecInfer | MCSS | MTAD |
| **HumanEval** | | | | | | | |
| **Llama-3-Instruct** | | | | | | | |
| *tokens/s* ↑ | 17.4 | 21.7 | 22.2 | 23.8 | 22.8 | 23.7 | **24.8** |
| *J/token* ↓ | 10.0 | 8.1 | 7.8 | 7.8 | 7.9 | 7.8 | **7.6** |
| *pass@1* ↑ | 37.8 | 35.9 | 32.9 | 32.9 | 31.0 | 32.0 | **38.4** |
| **Llama-3** | | | | | | | |
| *tokens/s* ↑ | 19.6 | 22.5 | 22.2 | 24.4 | 22.5 | 23.8 | **25.6** |
| *J/token* ↓ | 9.7 | 8.9 | 8.9 | 8.9 | 8.1 | 7.9 | **7.6** |
| *pass@1* ↑ | 19.5 | 20.0 | 15.9 | 16.0 | 17.7 | 17.0 | **22.0** |
| **Spider** | | | | | | | |
| **Llama-3-Instruct** | | | | | | | |
| *tokens/s* ↑ | 20.1 | 22.3 | 19.6 | 22.4 | 21.1 | 21.7 | **23.5** |
| *J/token* ↓ | 10.2 | 9.5 | 10.5 | 9.6 | 10.2 | 10.0 | **9.2** |
| *Acc* ↑ | 35.0 | 42.0 | 36.0 | 35.5 | 37.0 | 35.0 | **44.0** |
| **Llama-3** | | | | | | | |
| *tokens/s* ↑ | 23.3 | 32.3 | 31.1 | 32.1 | 32.6 | 32.7 ↓ | **33.3** |
| *J/token* | 8.2 | 7.9 | 7.5 | **7.1** | 8.1 | 8.0 | 7.8 |
| *Acc* ↑ | 30.5 | 29.5 | 21.5 | 23.0 | 21.5 | 24.0 | **35.0** |
| **MT-Bench** | | | | | | | |
| **Llama-3-Instruct** | | | | | | | |
| *tokens/s* ↑ | 25.9 | 23.4 | 26.0 | 26.2 | 26.3 | 26.8 ↓ | **29.8** |
| *J/token* ↓ | 10.8 | 12.2 | 10.0 | 9.9 | 10.0 | 9.9 | **9.2** |
| *score* ↑ | 4.15 | 4.26 | 4.10 | 4.11 | 4.01 | 4.02 | **4.40** |
| **Llama-3** | | | | | | | |
| *tokens/s* ↑ | 24.5 | 22.3 | 24.1 | 24.5 | 24.5 | 25.7 | **28.2** |
| *J/token* ↓ | 11.5 | 12.4 | 11.0 | 11.6 | 11.7 | 11.1 | **10.0** |
| *score* ↑ | 3.41 | 3.24 | 3.39 | 3.41 | 3.35 | 3.36 | **3.75** |

## 6 RELATED WORK

**EFFICIENT DECODING INFERENCE.** There are extensive studies on improving large model inference efficiency. Well-known methods include model quantization (Frantar et al., 2022; Lin et al., 2023), model pruning (Gale et al., 2019; Sanh et al., 2020), and model distillation (Hinton et al., 2015). Despite achieving significant speed-ups, a common drawback of these methods is that they have to sacrifice the model's effectiveness.

Non-autoregressive decoding more closely resembles our work. It is first proposed by (Gu et al., 2017) to generate multiple tokens in parallel. That is, the model simultaneously predicts $p(x_{t+k}|x_{1:t})$ ($k = 1, 2, \ldots$). Subsequent studies further improved the performance of parallel decoding by incorporating additional information (Wang et al., 2019; Sun et al., 2019; Li et al., 2019) or employing additional iterations to refine predictions (Ghazvininejad et al., 2019; Lee et al., 2018; Guo et al., 2020). However, these works require continuous training of the model and generally either compromise the model effectiveness or require task-dependent techniques to attain a comparable performance (Kim et al., 2023).

**SPECULATIVE DECODING.** Speculative decoding was recently proposed in (Leviathan et al., 2023; Chen et al., 2023) as a way to accelerate LLM inference. Spectr (Sun et al., 2023) enhances speculative decoding by letting the small model generate multiple i.i.d. draft sequences. While speculative decoding and Spectr use the large model to verify all the tokens drafted by the small model, BiLD (Kim et al., 2023) only calls the large model when the probability output by the small model is below a pre-defined threshold $\tau_1$. The large model rejects a token if its negative log-likelihood is larger than threshold $\tau_2$. SpecInfer (Miao et al., 2023) utilizes one or multiple small models to generate a draft token tree to increase the average acceptance length for each iteration. MCSS (Yang et al., 2024) further strengthens SpecInfer via sampling without replacement. All these methods can be perceived as exact or approximate versions of sampling tokens from the conditional distribution $p(x_t|x_{<t})$. Therefore, their output perplexity is bounded by greedy decoding.

An orthogonal direction to boost speculative decoding is to improve the effectiveness of the small draft model. It is obvious that if more draft tokens are accepted, the overall inference speed will increase. BiLD (Kim et al., 2023) employs a model prediction alignment technique to better train the small model. Liu et al. (Liu et al., 2023) propose online speculative decoding to continually update the draft model based on observed input data. Instead, Rest (He et al., 2023) uses a retrieval model to produce draft tokens. An alternative way is to train additional heads in the large model to predict future tokens. Representative works include EAGLE (Li et al., 2024) and MEDUSA (Cai et al., 2024). Importantly, these works are orthogonal to speculative decoding techniques, including our proposed method. This orthogonality means that the improvements offered by more accurate draft tokens could be combined with our method for better effectiveness.

## 7 CONCLUSION

We introduce multi-token assisted decoding, a process that enhances output quality while improving time and energy efficiency A distinctive aspect of our work is the exploration of speculative decoding's impact on inference energy consumption, an often neglected area in existing studies. This research contributes not only a novel decoding approach but also valuable insights for optimizing LLM deployment in real-world applications where considerations of both quality and efficiency are crucial.

## 8 ACKNOWLEGEMENT

This work was partially supported by NSF grants 2211557, 1937599, 2119643, 2303037, NSF 2312501, SRC JUMP 2.0 PRISM Center, NASA, Okawa Foundation, Amazon Research, Snapchat, and the CDSC industrial partners (https://cdsc.ucla.edu/partners/). The authors would also like to thank Marci Baun for editing the paper.

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

# A PROOF

## A.1 PROOF OF THEOREM 3.2

*Proof.*

$$PPL(x_{1:\Gamma_N}) = \exp\left(-\frac{1}{\Gamma_N}\sum_{i=1}^{\Gamma_N}\log p(x_i|x_{1:i-1})\right)$$
$$= \exp\left(-\frac{N}{\Gamma_N}\frac{1}{N}\sum_{i=1}^{N}\log p(x_{\Gamma_{i-1}:\Gamma_i}|x_{1:\Gamma_{i-1}})\right) \tag{10}$$

When $N \to \infty$, $\frac{\Gamma_N}{N} \to \bar{\gamma}$, and $\frac{1}{N}\sum_{i=1}^{N}\log p(x_{\Gamma_{i-1}:\Gamma_i}|x_{1:\Gamma_{i-1}}) \to \mathbb{E}_{x_{1:t}\in\mathcal{X}}\sum_{\gamma}\sum_{x_{t+1:t+\gamma}}P(\gamma)\tilde{p}(x_{t+1:t+\gamma}|x_{1:t})\log p(x_{t+1:t+\gamma}|x_{1:t}) = \mathbb{E}_\gamma L_p(\gamma, \tilde{p})$ $\square$

## A.2 PROOF OF COROLLARY 3.3

*Proof.* For deterministic multi token sampling, $\tilde{p}_{multi} = \arg\max \circ p$, so we have

$$L_p(\gamma, \tilde{p}_{multi}) = E_{x_{1:t}\in\mathcal{X}}\max_{x_{t+1:t+\gamma}}\log p(x_{t+1:t+\gamma}|x_{1:t}) \tag{11}$$

Notice that deterministic greedy sampling can be seen as a special case of MJGD where $\tilde{p}_{single}(x_{t+1:t+\gamma}|x_{1:t}) = 1$ if and only if $x_{t+i} = \arg\max_x p(x|x_{1:t+i-1})$ for $i = 1, \ldots, \gamma$. Let $x^*_{t+1:t+\gamma}$ be the tokens generated by deterministic MJGD and let $x'_{t+1:t+\gamma}$ be the tokens generated by deterministic greedy decoding. For any fixed $\gamma$ and $x_{1:t}$, we have $\log p(x'_{t+1:t+\gamma}|x_{1:t}) \leq \max_{x_{t+1:t+\gamma}}\log p(x_{t+1:t+\gamma}|x_{1:t}) = \log p(x^*_{t+1:t+\gamma}|x_{1:t})$. Therefore, $L_p(\gamma, \tilde{p}_{single}) \leq L_p(\gamma, \tilde{p}_{multi})$. Then with Theorem 3.2, we know that the perplexity of greedy decoding will be higher. $\square$

## A.3 PROOF OF LEMMA 3.4

We first prove the following Lemma.

**Lemma A.1.** *Let $PPL_p$ and $PPL_q$ denote the perplexity of tokens under distribution $p$ and $q$. When $N \to \infty$, we have*

$$\frac{PPL_p(x_{1:\Gamma_N})}{PPL_q(x_{1:\Gamma_N})} \leq \tau^{-\frac{1}{\bar{\gamma}}} \tag{12}$$

*where $\tau$ is the verification threshold.*

*Proof.* In the $i$-th iteration, the first $\gamma_i - 1$ tokens are the accepted draft tokens and the last token is sampled from $p$. Based on our verification criteria, we know that for the accepted draft tokens, we have

$$\frac{p(x_{\Gamma_{i-1}+1:\Gamma_{i-1}+\gamma_i-1}|x_{1:\Gamma_{i-1}})}{q(x_{\Gamma_{i-1}+1:\Gamma_{i-1}+\gamma_i-1}|x_{1:\Gamma_{i-1}})} \geq \tau. \tag{13}$$

So,

$$\frac{p(x_{1:\Gamma_N})}{q(x_{1:\Gamma_N})} \geq \tau^N\prod_{i=1}^{N}\frac{p(x_{\Gamma_i}|x_{1:\Gamma_i-1})}{q(x_{\Gamma_i}|x_{1:\Gamma_i-1})} \tag{14}$$

Notice that

$$\left(\prod_{i=1}^{N}\frac{p(x_{\Gamma_i}|x_{1:\Gamma_i-1})}{q(x_{\Gamma_i}|x_{1:\Gamma_i-1})}\right)^{\frac{1}{N}} = \exp\left(\frac{1}{N}\sum_{i=1}^{N}\log\left(\frac{p(x_{\Gamma_i}|x_{1:\Gamma_i-1})}{q(x_{\Gamma_i}|x_{1:\Gamma_i-1})}\right)\right) \tag{15}$$

When $N \to \infty$, since the last token at each iteration is sampled from $p$, we have

$$\frac{1}{N}\sum_{i=1}^{N}\log\left(\frac{p(x_{\Gamma_i}|x_{1:\Gamma_i-1})}{q(x_{\Gamma_i}|x_{1:\Gamma_i-1})}\right) \to \mathbb{E}_p\log\left(\frac{p(x_{\Gamma_i}|x_{1:\Gamma_i-1})}{q(x_{\Gamma_i}|x_{1:\Gamma_i-1})}\right) = KL(p,q) \geq 0 \tag{16}$$

So

$$\left(\prod_{i=1}^{N} \frac{p(x_{\Gamma_i}|x_{1:\Gamma_i-1})}{q(x_{\Gamma_i}|x_{1:\Gamma_i-1})}\right)^{\frac{1}{N}} \geq 1 \tag{17}$$

Therefore,

$$\frac{p(x_{1:\Gamma_N})}{q(x_{1:\Gamma_N})} \geq \tau^N \tag{18}$$

Thus,

$$\frac{PPL_p(x_{1:\Gamma_N})}{PPL_q(x_{1:\Gamma_N})} = \left(\frac{p(x_{1:\Gamma_N})}{q(x_{1:\Gamma_N})}\right)^{-\frac{1}{\Gamma_N}} \leq \tau^{-\frac{N}{\Gamma_N}} \to \tau^{-\frac{1}{\bar{\gamma}}} \tag{19}$$

$\square$

Now, we prove Lemma 3.4.

*Proof.*

$$-\log PPL_q(x_{1:\Gamma_N}) = \frac{1}{\Gamma_N}\sum_{i=1}^{N}(\log q(x_{\Gamma_{i-1}+1:\Gamma_i-1}|x_{1:\Gamma_{i-1}}) + \log q(x_{\Gamma_i}|x_{1:\Gamma_i-1})) \tag{20}$$

When $N \to \infty$, since the first $\gamma_i - 1$ tokens are sampled with beam decoding, we have

$$\frac{1}{N}\sum_{i=1}^{N}\log q(x_{\Gamma_{i-1}+1:\Gamma_i-1}|x_{1:\Gamma_{i-1}})) \to \mathbb{E}_\gamma \mathbb{E}_{x_{1:t}\in\mathcal{X}}\log q(x_{t+1:t+\gamma-1}|x_{1:t})$$

$$\geq (1-\epsilon)\mathbb{E}_\gamma \mathbb{E}_{x_{1:\Gamma_{i-1}}\in\mathcal{X}}\max_{x_{\Gamma_{i-1}+1:\Gamma_i-1}}q(x_{\Gamma_{i-1}+1:\Gamma_i-1}|x_{1:\Gamma_{i-1}}))$$

$$= (1-\epsilon)\mathbb{E}_\gamma L_q(\gamma-1, \arg\max\circ q) \tag{21}$$

Since the last token at each iteration is sampled from $p$, we have

$$\frac{1}{N}\sum_{i=1}^{N}\log q(x_{\Gamma_i}|x_{1:\Gamma_i-1})) \to \mathbb{E}_{x_{1:t}\in\mathcal{X}}\mathbb{E}_p \log q(x_{t+1}|x_{1:t}) = -H(p,q) \tag{22}$$

So

$$-\log PPL_q(x_{1:\Gamma_N}) \geq \frac{1-\epsilon}{\bar{\gamma}}\mathbb{E}_{\gamma,x_{1:\Gamma_{i-1}}\in\mathcal{X}}\max_{x_{t+1:t+\gamma}}q(x_{t+1:t+\gamma}|x_{1:t})) - \frac{H(p,q)}{\bar{\gamma}} \tag{23}$$

$$PPL_q(x_{1:\Gamma_N}) \leq \exp\left(\frac{H(p,q)}{\bar{\gamma}} - \frac{1-\epsilon}{\bar{\gamma}}\mathbb{E}_\gamma L_q(\gamma-1, \arg\max\circ q)\right) \tag{24}$$

$\square$

## A.4 PROOF OF THEOREM 3.5

*Proof.* We have

$$\lim_{N\to\infty}\frac{PPL_p(x_{1:\Gamma_N})}{PPL_p(x^*_{1:\Gamma_N})} \leq \tau^{-\frac{1}{\bar{\gamma}}}\lim_{N\to\infty}\frac{PPL_q(x_{1:\Gamma_N})}{PPL_p(x^*_{1:\Gamma_N})} \quad (Lemma A.1)$$

$$= \tau^{-\frac{1}{\bar{\gamma}}}\frac{\lim_{N\to\infty}PPL_q(x_{1:\Gamma_N})}{\exp\left(-\frac{1}{\bar{\gamma}}\mathbb{E}_\gamma L_p(\gamma, \arg\max\circ p)\right)} \quad (Theorem 3.2)$$

$$\leq \tau^{-\frac{1}{\bar{\gamma}}}\frac{\exp\left(\frac{H(p,q)}{\bar{\gamma}} - \frac{1-\epsilon}{\bar{\gamma}}\mathbb{E}_\gamma L_q(\gamma-1, \arg\max\circ q)\right)}{\exp\left(-\frac{1}{\bar{\gamma}}\mathbb{E}_\gamma L_p(\gamma, \arg\max\circ p)\right)} \quad (Lemma 3.4)$$

$$= \tau^{-\frac{1}{\bar{\gamma}}}\exp\left(\frac{H(p,q)}{\bar{\gamma}} - \frac{1-\epsilon}{\bar{\gamma}}\mathbb{E}_\gamma L_q(\gamma-1, \arg\max\circ q) + \frac{1}{\bar{\gamma}}\mathbb{E}_\gamma L_p(\gamma, \arg\max\circ p)\right) \tag{25}$$

Notice that $L_p(\gamma, \arg\max \circ p) \geq L_p(\gamma + 1, \arg\max \circ p)$ for any $\gamma$. This is because for any $x_{1:t}$, $\max_{x_{t+1:t+\gamma}} \log p(x_{t+1:t+\gamma}|x_{1:t}) \geq \max_{x_{t+1:t+\gamma+1}} (\log p(x_{t+1:t+\gamma}|x_{1:t}) + \log p(x_{t+\gamma+1}|x_{1:t+\gamma})) = \max_{x_{t+1:t+\gamma+1}} \log p(x_{t+1:t+\gamma+1}|x_{1:t})$.

So

$$
\lim_{N\to\infty} \frac{PPL_p(x_{1:\Gamma_N})}{PPL_p(x^*_{1:\Gamma_N})}
$$
$$
\leq \tau^{-\frac{1}{\bar{\gamma}}} \exp\left( \frac{H(p,q)}{\bar{\gamma}} + \frac{\epsilon}{\bar{\gamma}} \mathbb{E}_\gamma L_p(\gamma, \arg\max \circ p) + \frac{1-\epsilon}{\bar{\gamma}} (\mathbb{E}_\gamma L_p(\gamma, \arg\max \circ p) - \mathbb{E}_\gamma L_q(\gamma, \arg\max \circ q)) \right)
$$
(26)

Since $\epsilon \leq 0$, and $L_p(\gamma, \arg\max \circ p)$ is the maximum log-likelihood, which is larger than the expected log-likelihood (i.e., negative entropy), we have

$$
\frac{\epsilon}{\bar{\gamma}} \mathbb{E}_\gamma L_p(\gamma, \arg\max \circ p)
$$
$$
= \frac{\epsilon}{\bar{\gamma}} E_\gamma \mathbb{E}_{x_{1:t} \in \mathcal{X}} \max_{x_{t+1:t+\gamma}} \log p(x_{t+1:t+\gamma}|x_{1:t})
$$
$$
\leq \frac{\epsilon}{\bar{\gamma}} E_\gamma \mathbb{E}_{x_{1:t} \in \mathcal{X}} \sum_{x_{t+1:t+\gamma}} p(x_{t+1:t+\gamma}|x_{1:t}) \log p(x_{t+1:t+\gamma}|x_{1:t})
$$
$$
= -\epsilon H(p)
$$
(27)

In addition

$$
\mathbb{E}_\gamma L_p(\gamma, \arg\max \circ p) - \mathbb{E}_\gamma L_q(\gamma, \arg\max \circ q)
$$
$$
= \mathbb{E}_\gamma (L_p(\gamma, \arg\max \circ p) - L_q(\gamma, \arg\max \circ q))
$$
$$
= \mathbb{E}_\gamma \left( \mathbb{E}_{x_{1:t} \in \mathcal{X}} \max_{x_{t+1:t+\gamma}} \log p(x_{t+1:t+\gamma}|x_{1:t}) - \mathbb{E}_{x_{1:t} \in \mathcal{X}} \max_{x_{t+1:t+\gamma}} \log q(x_{t+1:t+\gamma}|x_{1:t}) \right)
$$
$$
= \mathbb{E}_\gamma \mathbb{E}_{x_{1:t} \in \mathcal{X}} \left( \max_{x_{t+1:t+\gamma}} \log p(x_{t+1:t+\gamma}|x_{1:t}) - \max_{x_{t+1:t+\gamma}} \log q(x_{t+1:t+\gamma}|x_{1:t}) \right)
$$
$$
\leq \mathbb{E}_\gamma \mathbb{E}_{x_{1:t} \in \mathcal{X}} \max_{x_{t+1:t+\gamma}} (\log p(x_{t+1:t+\gamma}|x_{1:t}) - \log q(x_{t+1:t+\gamma}|x_{1:t}))
$$
$$
= \mathbb{E}_\gamma \mathbb{E}_{x_{1:t} \in \mathcal{X}} \max_{x_{t+1:t+\gamma}} \left( \sum_{i=1}^{\gamma} \log p(x_{t+i}|x_{1:t+i-1}) - \log q(x_{t+i}|x_{1:t+i-1}) \right)
$$
$$
\leq \mathbb{E}_\gamma \mathbb{E}_{x_{1:t} \in \mathcal{X}} U\gamma \quad (because \; \| \log p(x|x_{1:t}) - \log q(x|x_{1:t}) \|_\infty \leq U)
$$
$$
= U\bar{\gamma}
$$
(28)

And $H(p,q) = H(p) + KL(p\|q)$.

$$
KL(p\|q) = \mathbb{E}_{x_{1:t} \in \mathcal{X}} \sum_x p(x|x_{1:t})(\log p(x|x_{1:t}) - \log p(x|x_{1:t}))
$$
$$
\leq \mathbb{E}_{x_{1:t} \in \mathcal{X}} \sum_x p(x|x_{1:t}) U \leq U
$$
(29)

So $H(p,q) \leq H(p) + U$. Therefore,

$$
\lim_{N\to\infty} \frac{PPL_p(x_{1:\Gamma_N})}{PPL_p(x^*_{1:\Gamma_N})} \leq \tau^{-\frac{1}{\bar{\gamma}}} \exp\left( \frac{(1-\epsilon\bar{\gamma})H(p) + (1-\epsilon+\bar{\gamma})U}{\bar{\gamma}} \right)
$$
(30)

$\square$

## A.5 PROOF OF THEOREM 3.6

*Proof.* Recall that we accept $x_{t+1:t+j}$ if and only if $\log p(x_{t+1:t+j}|x_{1:t}) - \log q(x_{t+1:t+j}|x_{1:t}) \geq \log \tau$. Since $\| \log p(x|x_{1:t}) - \log q(x|x_{1:t}) \|_\infty \leq U$, we have

$$
\log p(x_{t+1:t+j}|x_{1:t}) - \log q(x_{t+1:t+j}|x_{1:t}) \geq -jU
$$
(31)

Therefore $x_{t+1:t+j}$ is always accepted if $j \leq \frac{|\log \tau|}{U}$. So $\bar{\gamma} \geq \frac{|\log \tau|}{U}$ $\square$

# B  PSEUDOCODE OF MTAD

See Algorithm 1.

---

**Algorithm 1** One Iteration of MTAD Algorithm

---

1: **Input:** draft model $M_q$, target model $M_p$, $input$, threshold $\tau$
2:                               # Sample draft sequences from $M_q$ with beam sample.
3: $\boldsymbol{x}, \boldsymbol{q} \leftarrow \text{beamSample}(M_q, input)$       # $\boldsymbol{x}_i$ is the $i$-th draft token. $\boldsymbol{q}_i = q(\boldsymbol{x}_{1:i}|input)$
4: $\boldsymbol{P} \leftarrow M_p(input, \boldsymbol{X})$           # $\boldsymbol{P} \in \mathbf{R}^{(\gamma+1)\times|V|}$, $\boldsymbol{P}_{i,j} = p(x = j|\boldsymbol{x}_{1:i-1}, input)$
5:                               # Select the longest accepted draft sequence
6: $p \leftarrow 1, \eta \leftarrow -1$
7: **for** $i = 1$ **to** $\gamma$ **do**
8:      $j \leftarrow \boldsymbol{x}_i$
9:      $p \leftarrow p * \boldsymbol{P}_{i,j}, q \leftarrow \boldsymbol{q}_i$
10:      **if** $\tau < \min(1, \frac{p}{q})$ **then**
11:          $\eta \leftarrow j$                        # longest accepted prefix so far
12:      **end if**
13: **end for**
14:                               # Sample the next token using results of $M_p$
15: $\boldsymbol{p}' \leftarrow P_{\eta+1}$
16: $t \sim \boldsymbol{p}'$
17: **return** $[\boldsymbol{x}_1, \ldots, \boldsymbol{x}_\eta, t]$

---

Table 6: Dataset Statistics

| Dataset | Task | Avg. Input Len |
|---|---|---|
| ChatGPT-Prompt | Instruction | 25.2 |
| ChatAlpaca | Chat | 277.7 |
| CNNDM | Summarization | 3,967.1 |
| Spider | Text-to-SQL | 347.68 |
| MT-Bench | Various[1] | N/A[2] |
| HumanEval | Coding | 67 |

# C  ADDITIONAL EXPERIMENTS

## C.1  ADDITIONAL DATASETS AND MODEL FAMILY

Here we report the additional experiment results with three more datasets: (1) ChatGPT-Prompt (Rashad, 2023), (2) ChatAlpaca (Bian et al., 2023), (3) CNN Dailymail (See et al., 2017). We use two public LLM families in our experiments: OPT (Zhang et al., 2022) and Llama-2 (Touvron et al., 2023). We set the large model to be OPT-13B and Llama-2-13B as they are the largest models that can run on a single 40GB GPU, and utilize Llama-68M (Miao et al., 2023) and OPT-125M as the small models.

Table 7 shows the full evaluation results, and Table 9 displays the downstream performance. Table 8 depicts the average number of generated token per iteration for different algorithms. The experiment results demonstrate that MTAD achieves better efficiency, better perplexity, as well as better downstream performance.

Table 7: Inference efficiency and output perplexity of different methods on ChatGPT-Prompt (CP), ChatAlpaca (CA), CNNDailyMail (CD), Spider (SP), and MT-Bench (MT) datasets. **Bold numbers** mark the best result, underlined numbers mark the second best.

| | | | SpD | BiLD | Spectr | SpecInfer | MTAD |
|---|---|---|---|---|---|---|---|
| CP | Llama-2 | speed (token/s) ↑ | 36.8±0.53 | 34.4±0.87 | 45.1±1.32 | 29.7± 0.40 | **63.0±0.20** |
| | | energy (J/token) ↓ | 6.62±0.91 | 7.45±0.90 | 5.17±0.88 | 9.52±0.10 | **3.38±0.02** |
| | | perplexity ↓ | 3.64±0.11 | 3.15±0.06 | 3.64±0.08 | 3.64±0.11 | **2.06±0.06** |
| | OPT | speed (token/s) ↑ | 33.8±2.47 | 31.5±1.87 | 38.0±2.20 | 32.8± 0.58 | **55.8±0.30** |
| | | energy (J/token) ↓ | 7.48±0.07 | 8.75±0.13 | 6.08±0.11 | 10.3±1.49 | **3.61±0.03** |
| | | perplexity ↓ | 5.47±0.11 | 4.51±0.09 | 5.27±0.09 | 5.12±0.01 | **3.00±0.09** |
| CA | Llama-2 | speed (token/s) ↑ | 31.6±0.35 | 28.8±0.20 | 27.7±0.29 | 26.5±0.49 | **44.1±0.25** |
| | | energy (J/token) ↓ | 6.98±0.15 | 7.99±0.15 | 7.20±0.08 | 7.52±0.32 | **4.72±0.03** |
| | | perplexity ↓ | 2.13±0.03 | 1.95±0.03 | 2.15±0.01 | 2.15±0.01 | **1.88±0.05** |
| | OPT | speed (token/s) ↑ | 35.6±0.45 | 38.5±0.93 | 28.4±0.34 | 31.4±0.39 | **49.6±0.42** |
| | | energy (J/token) ↓ | 5.74±0.11 | 5.12±0.06 | 6.24±0.11 | 8.68±1.83 | **4.03±0.02** |
| | | perplexity ↓ | 3.32±0.10 | 2.60±0.06 | 3.16±0.06 | 3.42±0.03 | **2.07±0.03** |
| CD | Llama-2 | speed (token/s) ↑ | 30.7±0.18 | 30.5±0.21 | 25.0±0.31 | 24.6±0.06 | **44.2±0.99** |
| | | energy (J/token) ↓ | 7.07±0.19 | 7.41±0.16 | 8.22±0.19 | 7.59±0.85 | **4.80±0.12** |
| | | perplexity ↓ | 2.87±0.08 | 2.93±0.03 | 3.06±0.11 | 2.92±0.09 | **2.63±0.10** |
| | OPT | speed (token/s) ↑ | 31.7±0.91 | 30.9±0.80 | 23.7±0.40 | 25.7±0.36 | **43.6±0.33** |
| | | energy (J/token) ↓ | 6.37±0.11 | 6.71±0.17 | 7.31±0.17 | 8.03±0.63 | **4.86±0.03** |
| | | perplexity ↓ | 3.97±0.06 | 3.74±0.09 | 4.04±0.07 | 3.92± 0.34 | **3.17±0.06** |
| SP | Llama-2 | speed (token/s) ↑ | 24.0±0.28 | 26.2±0.08 | 24.2±0.29 | 23.8±0.20 | **26.4±0.28** |
| | | energy (J/token) ↓ | 10.75±0.02 | 9.84±0.07 | 11.0±0.08 | 11.0±0.76 | **9.01±0.07** |
| | | perplexity ↓ | 2.26±0.01 | 2.13±0.03 | 2.29±0.04 | 2.29±0.03 | **1.87±0.03** |
| | OPT | speed (token/s) ↑ | 24.6±0.30 | 29.9±0.55 | 19.8±0.13 | 24.1±0.10 | **34.4±0.46** |
| | | energy (J/token) ↓ | 15.6±3.55 | 13.6±3.07 | 20.1±2.52 | 16.9±2.75 | **11.7±2.36** |
| | | perplexity ↓ | 2.30±0.07 | 1.90±0.01 | 2.20±0.09 | 2.21± 0.01 | **1.63±0.03** |
| MT | Llama-2 | speed (token/s) ↑ | 23.0±1.10 | 23.7±1.43 | 19.1±2.71 | 23.7±2.03 | **29.4±2.71** |
| | | energy (J/token) ↓ | 7.99±0.26 | 7.40±0.19 | 9.27±0.54 | 9.20±0.73 | **6.71±1.19** |
| | | perplexity ↓ | 3.64±0.51 | 3.44±0.76 | 3.64±0.51 | 3.63±0.50 | **2.21±0.18** |
| | OPT | speed (token/s) ↑ | 34.0±3.00 | 44.7±2.92 | 28.7±2.46 | 28.5±2.74 | **48.0±1.80** |
| | | energy (J/token) ↓ | 12.1±0.36 | 6.23±0.67 | 12.9±1.73 | 13.2±1.88 | **6.11±0.82** |
| | | perplexity ↓ | 2.02±0.40 | 1.50±0.27 | 1.97±0.38 | 1.99± 0.33 | **1.10±0.03** |

Table 8: Average number of tokens generated at each iteration across all datasets.

| | Llama-2 | OPT |
|---|---|---|
| SpD | 2.02±0.05 | 2.60±0.06 |
| BiLD | 1.83±0.10 | 2.68±0.36 |
| Spectr | 2.73±0.43 | 3.45±0.42 |
| SpecInfer | 2.74±0.46 | 3.45±0.40 |
| MTAD | **3.17±0.43** | **4.30±0.03** |

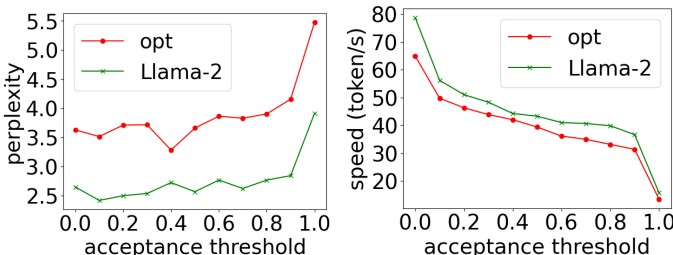

Figure 3: Effect of acceptance threshold on output perplexity and decoding speed.

Table 9: Downstream task scores of speculative decoding and MTAD. All the scores are higher the better.

|    |            | SpD  | MTAD |
|----|------------|------|------|
| CD | Rouge-L    | 0.114 | **0.118** |
| SP | EA         | 11.5 | **13.0** |
| MT | Humanities | 2.95 | **3.15** |
|    | Extraction | 1.80 | **2.50** |
|    | Roleplay   | 3.10 | **3.80** |
|    | Math       | **1.10** | 1.00 |
|    | Coding     | **1.25** | 1.10 |
|    | Reasoning  | **3.80** | 3.15 |
|    | STEM       | 2.85 | **3.10** |
|    | Writing    | **3.80** | 3.65 |
|    | Average    | 2.58 | **2.68** |

Table 10: Number of Generated Tokens Per Iteration Across Thresholds

|                 | Threshold | | | | |
|-----------------|------|------|------|------|------|
|                 | 0.1  | 0.3  | 0.5  | 0.7  | 0.9  |
| Llama-3         | 4.72 | 4.43 | 4.12 | 3.93 | 3.64 |
| Llama-3-Instruct | 4.74 | 4.51 | 4.28 | 4.06 | 3.79 |

## C.2 ABLATION STUDIES ON ACCEPTANCE THRESHOLD

Next, we evaluate the effect of acceptance threshold $\tau$. Intuitively, when we increase $\tau$ from 0 to 1, the acceptance criterion becomes stricter, the efficiency drops while the output perplexity increases. Surprisingly, this expectation is only partially correct. As shown in Figure 3, the efficiency indeed drops when $\tau$ increases. However, the perplexity rises when $\tau$ is close to 1. When $\tau = 1$, all the draft tokens are rejected, which makes MTAD equivalent to multinomial sampling. Similarly, when $\tau$ is close to 1, the advantage of multi-token joint decoding on effectiveness disappears, hence MTAD's perplexity is similar to the perplexity of multinomial sampling. Moreover, when $\tau$ ranges from 0.1 to 0.9, the performance of MTAD is relatively stable, suggesting that MTAD is not sensitive to the acceptance threshold.

In addition, Table 10 exhibits how the number of accepted tokens change with thresholds. We can see that the number of accepted tokens grows smaller when the threshold increases, which explains the speed degradation.

## C.3 ABLATION STUDIES ON NUMBER OF BEAMS

In Table 11, we investigate how the number of beams used in the beam decoding of the small model affects the inference performance. Increasing the number of beams boosts the quality of the draft tokens, which not only improves the output perplexity but also increases the average acceptance length and hence leads to better efficiency. But we can see that the increment slows down when the number of beams is too large. In addition, when the number of beams is too large, the inference cost of the small model will become too high.

---

[1]The tasks of MT-Bench cover humanities, extraction, roleplay, math, coding, reasoning, stem, writing, and STEM.

[2]MT-Bench contains multi-turn tasks where the input includes the responses of LLMs, so the input length is not fixed.

Table 11: Effect of beam count on inference performance (ChatGPT-Prompts dataset). Arrows indicate optimal direction (↑=higher better, ↓=lower better).

|  |  | \multicolumn{4}{c|}{Number of Beams} |
|  |  | 2 | 4 | 6 | 8 |
|---|---|---|---|---|---|
| Llama-2 | Speed (tok/s) ↑ | 55.9 | 59.9 | 60.2 | 61.3 |
|  | Energy (J/tok) ↓ | 2.43 | 2.25 | 2.22 | 2.20 |
|  | Perplexity ↓ | 2.44 | 2.12 | 2.14 | 2.10 |
| OPT | Speed (tok/s) ↑ | 51.0 | 54.1 | 54.3 | 55.9 |
|  | Energy (J/tok) ↓ | 2.50 | 2.32 | 2.36 | 2.30 |
|  | Perplexity ↓ | 3.63 | 3.16 | 3.42 | 3.19 |

## C.4    ABLATION STUDY OF TOP-K AND TOP-P SAMPLING

Table 12 demonstrates how the value of k and p in top-k and top-p warping affects our method. We can see that by changing the value of $k$ and $p$, MTAD consistently performs significantly better.

Table 12: Ablation study of $k$ and $p$ in top-k and top-p sampling

| K | P | \multicolumn{2}{c|}{Multinomial} | \multicolumn{2}{c|}{SpD} | \multicolumn{2}{c}{MTAD} |
|  |  | PPL | Tokens/sec | PPL | Tokens/sec | PPL | Tokens/sec |
|---|---|---|---|---|---|---|---|
| 20 | 0.9 | 3.74 | 22.6 | 3.64 | 36.8 | 2.06 | 63.0 |
| 20 | 0.8 | 3.06 | 22.7 | 3.10 | 38.5 | 1.93 | 58.8 |
| 10 | 0.9 | 3.03 | 22.7 | 3.22 | 38.5 | 1.95 | 62.5 |
| 10 | 0.8 | 2.56 | 22.7 | 2.53 | 40.0 | 1.80 | 62.5 |

## C.5    RESUTLTS WITH OPT-30B AND LLAMA-2-70B

Here we report the performances of different methods for OPT (350M and 30B) and Llama-2-Chat (7B and 70B). Table 13 shows the average performances across all datasets. MTAD always realizes the lowest perplexity and the best efficiency.

Table 13: Inference efficiency and output perplexity of different methods with OPT (350M,30B) and Llama-2-Chat (7B,70B). The mean and standard deviation are computed across all datasets. **Bold numbers** mark the best result, underlined numbers mark the second best.

|  |  | SpD | BiLD | Spectr | SpecInfer | MTAD |
|---|---|---|---|---|---|---|
| Llama-2 | speed (token/s) ↑ | 8.37±3.07 | 8.64±3.50 | 9.11±3.03 | 8.87±2.82 | **9.53±3.29** |
|  | energy (J/token) ↓ | 138±87.7 | 142±99.7 | 122±66.4 | 125±65.4 | **119±67.7** |
|  | perplexity ↓ | 1.77±0.22 | 1.69±0.25 | 1.73±0.24 | 1.73±0.24 | **1.52±0.19** |
| OPT | speed (token/s) ↑ | 15.3±1.64 | 14.5±1.96 | 17.0±4.14 | 17.4±4.00 | **19.5±4.11** |
|  | energy (J/token) ↓ | 72.4±11.5 | 79.6±3.03 | 68.2±16.7 | 62.4±10.3 | **60.0±12.8** |
|  | perplexity ↓ | 4.74±1.96 | 3.50±1.42 | 4.55±1.93 | 4.49± 1.95 | **2.74±0.87** |

## C.6    ADDITIONAL EXPERIMENTS ON CNNDM AND SPIDER

Table 14 and Table 15 reveal the downstream effectiveness of our method on CNNDM and Spider when the small model and large model are fine-tuned on the dataset. We can see that MTAD is consistently more effective and has a faster decoding speed.

Table 14: Comparison of ROUGE-L Scores and Tokens per Second under Different Fine-Tuning Conditions on CNNDM

| Method | No Fine-Tune | | Fine-Tune 68M | | Fine-Tune Both | |
|---|---|---|---|---|---|---|
| | ROUGE-L | Tokens/sec | ROUGE-L | Tokens/sec | ROUGE-L | Tokens/sec |
| SpeD | 0.114 | 37.7 | 0.114 | 20.4 | 0.164 | 24.3 |
| MTAD | 0.118 | 44.2 | 0.121 | 25.0 | 0.168 | 27.1 |

Table 15: Comparison of Execution Accuracy (EA) and Tokens per Second under Different Fine-Tuning Conditions on Spider

| Method | No Fine-Tune | | Fine-Tune 68M | | Fine-Tune Both | |
|---|---|---|---|---|---|---|
| | EA | Tokens/sec | EA | Tokens/sec | EA | Tokens/sec |
| SpD | 11.5 | 28.5 | 11.5 | 27.8 | 16.3 | 25.6 |
| MTAD | 13.0 | 30.3 | 14.8 | 32.3 | 18.3 | 29.4 |

## C.7 VISUALIZATION OF PERPLEXITY AND OUTPUT QUALITY

To further illustrate the relationship between perplexity and downstream performance, we present a scatter plot in Figure 4. The plot depicts the correlation between relative downstream scores (normalized by the score of multinomial sampling) and relative perplexity (normalized by the perplexity of multinomial sampling) across 7 decoding algorithms, 3 datasets, and 2 model configurations. The results confirm that lower perplexity generally correlates with higher output quality.

## C.8 CORRELATION BETWEEN ENERGY AND SPEED

Figure 5 shows there is a correlation between speed and energy, whether we consider the entire table or focus on a specific dataset and model. For fairness, all methods for a given dataset and model were run on the same machine nodes. However, for a fixed method (e.g., Spectr), experiments on different datasets and models might be conducted on different nodes (all equipped with L40 GPUs). We did notice that the same configuration run on different machines may have varied energy consumption. This variation introduces some randomness, which could make the correlation appear less consistent across datasets and models.

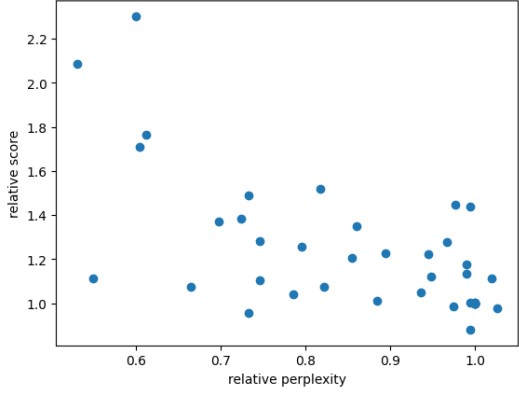

Figure 4: Relationship between relative perplexity (normalized by multinomial sampling's perplexity) and relative performance score (normalized by multinomial sampling's score).

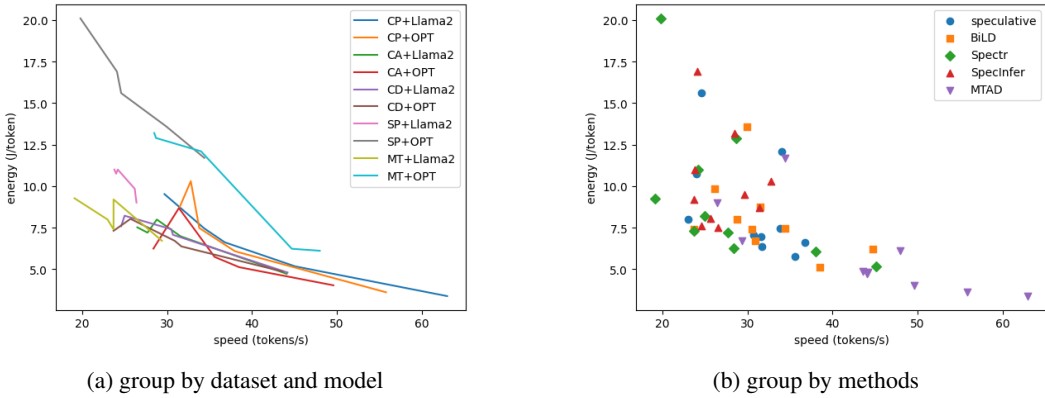

(a) group by dataset and model          (b) group by methods

Figure 5: Correlation between speed and energy

## D   ENERGY CONSUMPTION MEASUREMENT

To get GPU power every second, we run the command "`nvidia-smi –query-gpu=power.draw –format=csv`". We add the results up to determine the total energy consumption. We use average energy consumption per token to measure energy efficiency. There is a recent study pointing out the measurement error using `nvidia-smi` (Yang et al., 2023). We follow the three principles proposed in (Yang et al., 2023) to minimize this error.

## E   CONFIGURATION

The experiments are conducted on a machine with 1 Nvidia L40 GPU (48 GB), 4 CPUs, and 50 GB main memory, using a batch size of 1, which is common for online serving (Schuster et al., 2022). We set the maximum running time to be an hour for each baseline. We use average tokens/second to measure the inference speed and use average energy consumption per token to measure energy efficiency.

## F   HYPER-PARAMETER DETAILS

In the experiments, we follow the settings in (Bear, 2024) to warp the sampling distribution $p$ and $q$ with the following steps, which are the default warping operations in a public speculative decoding implementation. Specifically, we first keep the probabilities of top 10 tokens unchanged, and set the probabilities of other tokens to 0, then normalize the distribution. Then we sort the tokens based on their distributions in descending order and keep the first $K$ tokens such that their cumulative probabilities is larger than 0.9, while set the probabilities of other tokens to 0.

For different methods, we choose their hyper-parameters based on a small validation set. We select the set of hyper-parameters that make the corresponding method have best output perplexity. For MTAD, we choose the beam width from $\{4, 8\}$, the number of draft tokens from $\{3, 4\}$, and the acceptance threshold from $\{0.1, 0.3, 0.5, 0.7, 0.9\}$.

## G   LICENSE OF DATASETS AND MODELS

Datasets:

- ChatGPT-Prompts: Non (https://huggingface.co/datasets/MohamedRashad/ChatGPT-prompts)
- ChatAlpaca: Apache-2.0 License
- CNN Dailymail: Apache-2.0 License
- MTBench: Apache-2.0 License

- Spider: CC BY-SA 4.0 License
- HumanEval: MIT License

Models

- OPT-125M and OPT-13B: Model License (`https://github.com/facebookresearch/metaseq/blob/main/projects/OPT/MODEL_LICENSE.md`)
- Llama-68M: Apache-2.0 License
- Llama-2-13B: Llama-2 Community License Agreement
- Llama-3: the Llama 3.1/3.2 Community License

Codes

- LLMSpeculativeSampling (`https://github.com/feifeibear/LLMSpeculativeSampling`)

