# OpenReview forum: "Optimized Multi-Token Joint Decoding With Auxiliary Model for LLM Inference"
_ICLR.cc/2025/Conference — ICLR 2025 Poster_

### Official Review · Reviewer_ds8N · 2024-10-29

**Soundness:** 3
**Presentation:** 3
**Contribution:** 3
**Rating:** 6
**Confidence:** 3

**Summary:**

Considering the inference efficiency, this paper proposes multi-token joint decoding (MTJD), which generates multiple tokens from their joint distribution at each iteration. More importantly, this paper theoretically proves that their method can reduce perplexity and improve task performance compared to single token decoding. Besides, to mitigate the high cost of sampling from the joint distribution of multiple tokens, it introduce a small model to assist the decoding of a larger model. Empirical evaluations reveal that the method reduces perplexity
by 21.2% and improves downstream performance compared to standard single-token sampling. It also achieves a 1.42× speed-up and consumes 1.54× less energy than conventional speculative decoding methods.

**Strengths:**

1. This paper theoretically proves that multi-token joint decoding (MTJD) can reduce perplexity and improve task performance compared to single token decoding.
2. This paper introduces multi-token assisted decoding (MTAD), a novel framework designed to accelerate multi-token joint decoding. Theoretically, this paper demonstrate that MTAD closely approximates exact MTJD with bounded error.
3. These results highlight MTAD’s ability to make multi-token joint decoding both effective and efficient.

**Weaknesses:**

1. This paper primarily conducted experiments on the OPT series and Llama 2 series models. However, the performance of these large models has already fallen far behind the recent Llama 3.1 and Llama 3.2. Could you provide more results on the latest models to demonstrate the generality of the results?

2. The previous methods compared in this paper were all published last year. It would be beneficial to compare with some recently published methods, such as MEDUSA-2 and "Better & Faster Large Language Models via Multi-token Prediction", to prove the effectiveness of the results.

3. In this paper, a major point is that multi-token joint decoding can reduce perplexity and improve task performance. However, the experimental results do not reflect related analytical experiments. Adding relevant experiments could enhance the self-consistency of the method.

**Questions:**

Please refer to the weakness.

---

> ### Author Response · Authors · 2024-11-21
> **Response to Reviewer ds8N**
>
> We sincerely appreciate the reviewer’s insightful suggestions and we hope our responses can address your concerns.
>
> # Results with Llama-3 Models
> To address your concern about the results with newer models, we have added a more comprehensive evaluation using Llama-3 series models, with 8B (8B-Instruct) as the target model and 1B (1B-Instruct) as the draft model. These results are detailed in the "Common Reply on Perplexity and Downstream Performance" and demonstrate the robustness and generality of our proposed method across state-of-the-art models.
>
> # Comparisons with Recently Published Methods
> We have also incorporated additional baselines for comparison. Specifically, we include typical decoding from MEDUSA-2 as an additional baseline. Regarding "Better & Faster Large Language Models via Multi-token Prediction," this paper proposes new model architecture to generate draft tokens while using existing speculative decoding algorithms for inference. So it is orthogonal to our work, and our method is applicable to its new model architecture. **Notably, many recently published papers in this area focus on improving draft models rather than introducing new decoding algorithms.** For example, EAGLE-2 [1] proposes a new model architecture to generate draft tokens, but still use SpecInfer as the decoding method. These works are orthogonal to our paper. **To our knowledge,**  **the baselines used in our paper and rebuttal are still state-of-the-art speculative decoding algorithms.**
>
> # Experiments with MTJD
> We also acknowledge the importance of providing analytical experiments to support our claims about multi-token joint decoding (MTJD). As suggested, we report the output quality of MTJD and multinomial sampling on benchmarks such as Spider, MTBench, and HumanEval (Pass@1) in the "Common Reply on Perplexity and Downstream Performance." These results further validate the effectiveness of our method in reducing perplexity and improving task performance.
>
> We hope these additional experiments and clarifications address your concerns and demonstrate the robustness, effectiveness, and self-consistency of our work. Please feel free to refer to the common response for further details, and let us know if additional information is needed.
>
> [1] Eagle-2: Faster inference of language models with dynamic draft trees

---

> > ### Author Response · Authors · 2024-11-24
> > **Gentle Reminder: Awaiting Your Response to Our Rebuttal**
> >
> > Dear Reviewer ds8N,
> >
> > I hope this message finds you well. We recently submitted our rebuttal and would like to kindly request your feedback on our responses.
> >
> > We understand that your schedule is demanding and greatly appreciate the time and effort you dedicate to the review process. Your insights are invaluable to us, and we are eager to address any further questions or concerns you may have.
> >
> > Thank you for your attention to this matter. We look forward to your response.
> >
> > Best regards,
> >
> > Authors

---

> > > ### Comment · Reviewer_ds8N · 2024-11-25
> > > **Response to Authors**
> > >
> > > Thanks for your response. I appreciate your effort to conduct experiments to address my concerns and will raise my score.

---

> > > > ### Author Response · Authors · 2024-11-25
> > > >
> > > > Thank you very much for your kind response and for taking the time to review our additional experiments and explanations. We sincerely appreciate your thoughtful feedback and your willingness to raise your score. Your insights and suggestions have been invaluable in helping us strengthen our work.

---

### Official Review · Reviewer_v4L1 · 2024-11-03

**Soundness:** 2
**Presentation:** 3
**Contribution:** 2
**Rating:** 6
**Confidence:** 4

**Summary:**

This paper focuses on improving the inference speed of large language models (LLMs). The authors propose a variant of speculative sampling that enhances both inference speed and output quality. Specifically, they explore multi-token joint decoding (MTJD), which generates multiple tokens from their joint distribution at each iteration, theoretically reducing perplexity and improving task performance. Building on this, they incorporate speculative sampling to approximate multi-token joint decoding and design an efficient algorithm, multi-token assistant decoding (MTAD), which achieves bounded error with respect to MTJD. Empirical evaluations demonstrate the effectiveness of the proposed method in terms of speed and perplexity.

**Strengths:**

- The writing is good and easy to follow.
- The addressed problem is of great importance to the community.
- A new metric is provided to evaluate the effectiveness of speculative sampling, which is insightful.
- Strong experimental results:
  - Evaluated across a range of datasets
  - Shows significant improvements in terms of perplexity, speed, and energy efficiency.

**Weaknesses:**

- In line 114, the authors claim, “As shown in Table 1, lower perplexity correlates with improved downstream performance, even in one of today’s largest models.” This assertion is questionable. The results in Table 1 are based exclusively on experiments with the Spider dataset, but in many cases, lower perplexity does not inherently lead to better downstream performance. For example, in tasks like machine translation and abstractive summarization, a lower perplexity score sometimes fails to correlate with improved downstream performance. This also can be validated by the right plot in Figure 1.
- Lack of guarantee on quality: In vanilla speculative sampling and multi-draft speculative sampling, one of their advantages is the property of being lossless. This ensures that the quality of output is maintained, which further accelerates LLM inference. However, the proposed method does not have such guarantee.
- The verification process in this work is similar to that in [1], as both aim to verify a sub-sequence of tokens in a single step rather than one token at a time. However, this study does not include a comparison between MTAD and the method in [1]. The author should further compare their work with it to further improve the quality.
- The experiments also have some limitations:
  - This paper places too much emphasis on perplexity. Since a lower perplexity score does not always correlate with improved downstream performance, the authors should provide a more comprehensive evaluation of downstream performance. I noticed that Table 6 contains some comparisons on downstream performance, but this is insufficient. I encourage the authors to add a "downstream performance" metric to Table 4 so that readers can better understand the effectiveness of the proposed method.
  - I still do not fully understand why the proposed method significantly outperformed multi-draft speculative sampling methods such as Specter and SpecInfer. I suspect this is because (1) the selected baseline is weak, and (2) the configurations are not fair. For example, in Table 13, the number of sequences in Specter and SpecInfer does not match the number of beams in MTAD. I encourage the authors to do the following: (1) select more recent multi-draft speculative sampling methods for comparison, and (2) ensure consistent configurations.


[1] Sun, et al. Optimal Block-Level Draft Verification for Accelerating Speculative Decoding.

**Questions:**

See above.

---

> ### Author Response · Authors · 2024-11-21
> **Response to Reviewer v4L1**
>
> # Correlation between Perplexity and Output Quality
> We appreciate your concern regarding the relationship between lower perplexity and downstream performance. While we acknowledge that this correlation may vary across tasks, we provide strong evidence supporting it within the context of our work.
>
> 1. **Empirical Evidence:** As detailed in our "Common Reply on Perplexity and Downstream Performance," we compare multi-token joint decoding (MTJD, k=4) with single-token multinomial sampling across multiple datasets. MTJD consistently achieves significantly lower perplexity and better downstream performance. Furthermore, newly added Figure 4 in the Appendix C.4 illustrates a clear correlation between perplexity and downstream task performance across baselines, datasets, and models used in our experiments.
> 2. **Established Decoding Principles:** The connection between likelihood and quality is well-recognized in decoding strategies like beam search, which selects sequences with the highest likelihood (i.e., lowest perplexity). Previous works [4] confirm that beam search generally outperforms greedy decoding in terms of output quality, further validating the correlation.
> 3. **Analysis of Figure 1:** While perplexity values from different models (e.g., OPT and Llama-2) are not directly comparable due to varying likelihood functions, the trend of lower perplexity correlating with better performance holds true within each model family.
>
> We acknowledge that in some cases, lower perplexity may not correspond to improved performance, often due to limitations in the target model itself. However, our theoretical insights, experimental results, and the broader literature collectively demonstrate that lower perplexity generally correlates with better output quality and downstream performance for our method.
>
> # Comprehensive Evaluation of Downstream Performance
> We appreciate the suggestion to include additional downstream performance metrics. In response, we conducted a comprehensive evaluation using Llama-3 models (Llama-3-8B, Llama-3-1B, and their instruct variants) across Spider, HumanEval, and MT-Bench. These results, provided in the "Common Reply on Perplexity and Downstream Performance," demonstrate that MJAD improves downstream performance while achieving the fastest speed among all methods.
>
> # Advantage over Multi-Draft Speculative Sampling Methods
> We appreciate the reviewer’s concern and provide clarification on why MTAD significantly outperforms multi-draft speculative decoding methods like Spectr and SpecInfer.
>
> **Superior Design of MTAD**
>
> In Spectr and SpecInfer, despite having multiple draft sequences, for each draft sequence, the verification terminates as soon as a token is rejected. In contrast, when a token is rejected, MTAD’s verification continues to seek if future tokens can pass the verification. So as shown in Table 5, **it allows our method to get a higher acceptance length than the baselines**. Additionally, multi-draft methods increase the number of draft sequences to enhance acceptance rates, but this incurs substantial computational and memory overhead for target model verification. The tree-attention mechanism only alleviates this issue, but the extra overhead is still unavoidable. On the other hand, **MTAD only outputs one draft sequence**, so there is no extra overhead in verification. Based on these two factors, it is not surprising MTAD is more efficient.
>
> **Why our experiment configuration is fair.**
>
> (1) **Our baselines are SOTA.** Although there are some recent papers about speculative decoding, most of them focus on how to obtain a better draft model and they simply adopt the existing decoding algorithms. For example, EAGLE-2 [2] and Gloeckle et al [3] propose new model architectures to generate draft tokens, but still use SpecInfer or other existing decoding methods. These works are orthogonal to our paper. **To our knowledge,**  **the baselines used in our paper and rebuttal are still state-of-the-art speculative decoding algorithms.**
>
> (2) **The configuration is fair.** In MTAD, the number of draft sequences is always 1, regardless of the number of beams, because beam decoding only returns the beam with the highest likelihood. So a higher beam number only increases the computation cost of the small model, but not the target model. Meanwhile, if we increase the number of draft sequences in Spectr and SpecInfer, there are more sequences to verify for the target model, which is much more expensive than increasing the beam number of the small model. **The setting reported in Table 13 achieves best efficiency for Spectr and SpecInfer.** **We tried setting the number of sequences to match the number of beams as suggested by the reviewer, it causes out-of-memory errors.**

---

> ### Author Response · Authors · 2024-11-21
> **Response to Reviewer v4L1**
>
> # Lack of Quality Guarantee
>
> We agree that being “lossless” is an appealing feature of speculative decoding. However, while MTAD is "lossy," the loss in accuracy is measured against MTJD, a decoding paradigm that is both theoretically and empirically superior to multinomial sampling (see results in the "Common Reply on Perplexity and Downstream Performance").
>
> 1. **Different Design Objectives:** Unlike lossless methods that aim to replicate multinomial sampling, MTAD is designed to achieve higher output quality and efficiency by approximating MTJD.
> 2. **Theoretical Bounds:** We provide a theoretical bound on the approximation error between MTAD and MTJD, ensuring that MTAD closely adheres to the benefits of MTJD.
> 3. **Empirical Performance:** With MTAD's "starting point" already outperforming multinomial sampling, our results consistently demonstrate better output quality and faster speeds compared to lossless speculative decoding methods.
>
> This distinction in design objectives positions MTAD as a fundamentally different approach, focused on achieving superior efficiency and effectiveness.
>
> # Comparison with Method [1]
> We appreciate the reviewer for highlighting the relevance of method [1]. While it shares a similarity with MTAD in allowing verification to continue after token rejection, the objectives and methodologies differ significantly. Specifically, since [1] aims to have identical distribution as multinomial sampling, its verification scheme significantly differs from the original speculative decoding and our method.
>
> Since [1] is a preprint paper and its code is not open-sourced, we use the reported results in the paper for comparison. According to [1], it consistently achieves a **5–8%** speed improvement over vanilla speculative decoding while maintaining an identical sampling distribution. In contrast, as shown in Table 4 and the supplementary experiments, MTAD achieves a **1.42x** speedup on Llama-2 and OPT models, and a **1.16x** speedup on Llama-3 models, compared to vanilla speculative decoding. **Moreover, MTAD delivers higher output quality** due to its design focus on approximating multi-token joint decoding (MTJD) rather than adhering to multinomial sampling.
>
> These results highlight MTAD’s advantages in both efficiency and effectiveness, underscoring the benefits of its distinct design objectives.
>
> [1] Sun, et al. Optimal Block-Level Draft Verification for Accelerating Speculative Decoding.
>
> [2]Eagle-2: Faster inference of language models with dynamic draft trees
>
> [3] Better & Faster Large Language Models via Multi-token Prediction
>
> [4] A Thorough Examination of Decoding Methods in the Era of LLMs

---

> > ### Author Response · Authors · 2024-11-24
> > **Gentle Reminder: Awaiting Your Response to Our Rebuttal**
> >
> > Dear Reviewer v4L1,
> >
> > I hope this message finds you well. We recently submitted our rebuttal and would like to kindly request your feedback on our responses.
> >
> > We understand that your schedule is demanding and greatly appreciate the time and effort you dedicate to the review process. Your insights are invaluable to us, and we are eager to address any further questions or concerns you may have.
> >
> > Thank you for your attention to this matter. We look forward to your response.
> >
> > Best regards,
> >
> > Authors

---

> > > ### Comment · Reviewer_v4L1 · 2024-11-25
> > > **Response to Authors**
> > >
> > > Thanks very much for your detailed reply. I really appreciate your efforts in this.
> > >
> > > Although I still believe that, in some cases, there may not be a strong correlation between perplexity (PPL) and downstream performance, the additional experiments and evidence have significantly alleviated most of my concerns on this concern.
> > >
> > > My primary concern at present is the comparison methodology. The authors have stated that they used the with-replacement method to construct the draft token tree, which is not the state-of-the-art method. I strongly encourage the authors to conduct additional experiments using the without-replacement method.

---

> > > > ### Author Response · Authors · 2024-11-25
> > > > **Comparision with Spectr and SpecInfer (sampling without replacement)**
> > > >
> > > > Thank you for your reply. We are glad that our additional comprehensive evaluation has alleviated most of your concerns and demonstrated that our method can indeed improve output quality.
> > > >
> > > > In response to your suggestion, we implemented **Spectr\*** and **SpecInfer\***, which are versions of Spectr and SpecInfer where draft tokens are sampled without replacement. The results are shown in the table below. From the table, we observe that the effect of sampling with or without replacement is not significant. We believe this is because repeated draft tokens are relatively infrequent when sampling with replacement in our experiments.
> > > >
> > > > |  |  |  | spectr | spectr* | specinfer | specInfer* | MTAD |
> > > > | :---- | :---- | :---- | :---- | :---- | :---- | :---- | :---- |
> > > > | Spider | Llama-3-Instruct | tokens/s | 22.4 | 22.4 | 21.1 | 21.7 | **23.5** |
> > > > |  |  | J/token | 9.6 | 9.7 | 10.2 | 9.8 | **9.2** |
> > > > |  |  | Acc | 35.5 | 35.5 | 37.0 | 35.0 | **44.0** |
> > > > |  | Llama-3 | tokens/s | 32.1 | 32.8 | 32.6 | 32.6 | **33.3** |
> > > > |  |  | J/token | 7.1 | **7.0** | 8.1 | 7.9 | 7.8 |
> > > > |  |  | Acc | 23.0 | 22.0 | 21.5 | 23.0 | **35.0** |
> > > > | MT-Bench | Llama-3-Instruct | tokens/s | 26.2 | 26.3 | 26.3 | 26.7 | **29.8** |
> > > > |  |  | J/token | 9.9 | 10.1 | 10.0 | 9.8 | **9.2** |
> > > > |  |  | score | 4.11 | 4.11 | 4.01 | 4.05 | **4.40** |
> > > > |  | Llama-3 | tokens/s | 24.5 | 24.5 | 24.5 | 24.5 | **28.2** |
> > > > |  |  | J/token | 11.6 | 11.5 | 11.7 | 11.4 | **10.0** |
> > > > |  |  | score | 3.41 | 3.41 | 3.35 | 3.35 | **3.75** |
> > > > | HumanEval | Llama-3-Instruct | tokens/s | 23.8 | 23.9 | 22.8 | 22.7 | **24.8** |
> > > > |  |  | J/token | 7.8 | 7.8 | 7.9 | 7.9 | **7.6** |
> > > > |  |  | pass@1 | 32.9 | 31.0 | 31.0 | 31.0 | **38.4** |
> > > > |  | Llama-3 | tokens/s | 24.4 | 23.9 | 22.5 | 23.3 | **25.6** |
> > > > |  |  | J/token | 8.9 | 9.0 | 8.1 | 7.9 | **7.6** |
> > > > |  |  | pass@1 | 16.0 | 15.9 | 17.7 | 15.0 | **22.0** |
> > > >
> > > > In addition, we want to clarify some issues regarding sampling without replacement:
> > > >
> > > > In Spectr and SpecInfer, both methods assume that draft tokens are sampled **independently** to ensure the output token aligns with the target distribution (Theorem 4.2 in the SpecInfer paper and Theorem 2 in Spectr paper). **Sampling without replacement breaks this assumption because the probability of selecting a token is affected by previous selections.** While this deviation may not significantly impact output quality in practice, it does invalidate the assumption that each draft sample is drawn independently. As a result, the formal **losslessness guarantee** cannot be directly extended to the without-replacement case without revisiting and modifying the theoretical framework.
> > > >
> > > > We hope this explanation and additional results address your concerns, and we sincerely appreciate your thoughtful feedback and encouragement.

---

> > > > > ### Comment · Reviewer_v4L1 · 2024-11-25
> > > > > **Response to Authors**
> > > > >
> > > > > Thanks for your response. However, I believe there might be some misunderstanding regarding the experiments:
> > > > >
> > > > > 1. There is no need to implement Spectr using without-replacement sampling because this method assumes that draft tokens are sampled independently. Implementing Spectr with without-replacement sampling would break its theoretical guarantees.
> > > > >
> > > > > 2. However, SpecInfer can be extended to support without-replacement sampling without compromising theoretical lossless guarantees. For more details, you may refer to the Eagle implementation [1] or relevant studies such as [2]. The implementation requires only minor modifications to the existing codebase.
> > > > >
> > > > > [1] EAGLE: Speculative Sampling Requires Rethinking Feature Uncertainty.
> > > > > [2] Multi-Candidate Speculative Decoding.

---

> ### Author Response · Authors · 2024-11-25
> **Additional experiments: MCSS**
>
> Thank you for providing a more detailed reference and clarification regarding the experiments. Based on your suggestion, we implemented Multi-Candidate Speculative Sampling (MCSS) from [2] with sampling without replacement. The updated results are shown below and included in the table within the common response (a new "MCSS" column in the second table). While MCSS is slightly faster than SpecInfer, our method, MTAD, still outperforms it both in efficiency and output quality.
>
> Additionally, as discussed at the end of Section 3.2, multi-draft speculative sampling methods (e.g., Spectr, SpecInfer, MCSS) are compatible and orthogonal to our approach for the following reasons:
>
> - Extensibility of MTAD: MTAD can be easily extended to generate multiple draft sequences, making it compatible with multi-draft methods.
> - Distinct Design Objectives: Multi-draft methods aim to be lossless with respect to multinomial sampling, while our primary design objective is to approximate multi-token joint decoding (MTJD) to achieve superior output quality and efficiency.
>
> We hope these clarifications and updated results address your concerns. Please let us know if further explanation is needed.
>
>
> |  |  |  | Lossy Speculative Decoding |  | Lossless Speculative Decoding |  |  | | Ours |
> | :---- | :---- | :---- | :---- | :---- | :---- | :---- | :---- | :---- |:----|
> |  |  |  | BiLD | Typical (Medusa) | spec | spectr | specinfer|MCSS | MTAD |
> | Spider | Llama-3-Instruct | tokens/s | 20.1 | 22.3 | 19.6 | 22.4 | 21.1 | 21.7| **23.5** |
> |  |  | J/token | 10.2 | 9.5 | 10.5 | 9.6 | 10.2 |10.0 |**9.2**|
> |  |  | Acc | 35.0 | 42.0 | 36.0 | 35.5 | 37.0 |35.0|**44.0**|
> |  | Llama-3 | tokens/s | 23.3 | 32.3 | 31.1 | 32.1 | 32.6 |32.7| **33.3**|
> |  |  | J/token | 8.2 | 7.9 | 7.5 | **7.1** | 8.1 | 8.0|7.8 |
> |  |  | Acc | 30.5 | 29.5 | 21.5 | 23.0 | 21.5 |24.0| **35.0** |
> | MT-Behcn | Llama-3-Instruct | tokens/s | 25.9 | 23.4 | 26.0 | 26.2 | 26.3 |26.8| **29.8** |
> |  |  | J/token | 10.8 | 12.2 | 10.0 | 9.9 | 10.0 |9.9| **9.2** |
> |  |  | score | 4.15 | 4.26 | 4.1 | 4.11 | 4.01 |4.02| **4.40** |
> |  | Llama-3 | tokens/s | 24.5 | 22.3 | 24.1 | 24.5 | 24.5 |25.7| **28.2** |
> |  |  | J/token | 11.5 | 12.4 | 11.0 | 11.6 | 11.7 | 11.1|**10.0** |
> |  |  | score | 3.41 | 3.24 | 3.39 | 3.41 | 3.35 |3.36| **3.75** |
> | HumanEval | Llama-3-Instruct | tokens/s | 17.4 | 21.7 | 22.2 | 23.8 | 22.8 |23.7| **24.8** |
> |  |  | J/token | 10.0 | 8.1 | 7.8 | 7.8 | 7.9 |7.8| **7.6** |
> |  |  | pass@1 | 37.8 | 35.9 | 32.9 | 32.9 | 31.0 |32.0|**38.4**|
> |  | Llama-3 | tokens/s | 19.6 | 22.5 | 22.2 | 24.4 | 22.5 |23.8|**25.6**|
> |  |  | J/token | 9.7 | 8.9 | 8.9 | 8.9 | 8.1 |7.9|**7.6**|
> |  |  | pass@1 | 19.5 | 20.0 | 15.9 | 16.0 | 17.7 |17.0| **22.0** |

---

> > ### Comment · Reviewer_v4L1 · 2024-11-25
> > **Response to Authors**
> >
> > Thanks for coducting additional experiments about MCSS. I really appreciate your efforts.
> >
> > I have some questions about the experimental results:
> > 1) What is the temperature setting in the experiment?
> > 2) Typically, the efficiency of these method should follow the following patterns: MCSS > Spectr > Specinfer because a) Both Specinfer and Spectr assume draft tokens are sampled independently, but Spectr is designed as an approximation of the optimal solution to optimal transport. b) MCSS is under different assumtion that draft tokens are sampled dependently allowing it to generate a greater diversity of drafts, which increases the probability of acceptance. For instance, when sampling with replacement and generating 10 drafts, the effective number of unique drafts may degrade to fewer than 10.  **However, I cannot observe such patterns consistently in the experimental results. Can the authors give more explanations about this?**

---

> ### Author Response · Authors · 2024-11-25
>
> Thank you for your thoughtful questions and observations.
>
> > Temperature setting
>
> The temperature is set to 1 in the experiments above.
>
> > Efficiency Patterns: "MCSS > Spectr > SpecInfer"
>
> First, based on our experiment results, the average speed of MCSS is 25.73 tokens/s, the speed of spectr is 25.57 tokens/s, the speed of SpecInfer is 24.97 tokens/s. **These results align with the reviewer’s claim that MCSS demonstrates slightly higher efficiency, followed by Spectr and then SpecInfer.**
>
> Second, **the differences in speed between these methods are relatively small.** For example, Table 4 of the MCSS paper reports that MCSS without replacement is on average only **4%** faster than MCSS with replacement (equivalent to SpecInfer). On Llama-13 models—the most similar to the Llama-3 models used in our experiments—the speed up is just **1%**. In our experiments, MCSS is on average **3%** faster than SpecInfer, which is consistent with these findings. Besides, given these small differences, it is not surprising that Spectr can sometimes be faster than MCSS without replacement.
>
> Thrid, the assertion that "MCSS > Spectr > SpecInfer" appears to be an empirical observation rather than a theoretical guarantee. As suggested by the reviewer, sampling without replacement improves efficiency most greatly when the effective number of unique drafts is significantly smaller than the total number of draft sequences. This efficiency gain depends on the sampling distribution, which is affected by draft and target models, as well as the datasets used. Therefore, it is expected that MCSS without replacement is not always the fastest.
>
> We hope this explanation clarifies the observed results. Please feel free to reach out if you have further questions or need additional details.

---

> > ### Comment · Reviewer_v4L1 · 2024-11-25
> > **Response to Authors**
> >
> > Thanks a lot for the clarifications. Please include these experiments and discussions in the final version of your paper.
> >
> > I will update my ratings given these additional experiments and discussions.

---

> > > ### Author Response · Authors · 2024-11-25
> > >
> > > Thank you so much for your thoughtful feedback and for taking the time to review our additional experiments and discussions. We sincerely appreciate your valuable insights and are glad that the clarifications addressed your concerns. We will make sure to include these experiments and discussions in the final version of the paper. Thank you again for your support and for updating your ratings!

---

### Official Review · Reviewer_h4X4 · 2024-11-04

**Soundness:** 3
**Presentation:** 3
**Contribution:** 2
**Rating:** 6
**Confidence:** 4

**Summary:**

This paper introduces a new framework named multi-token assisted decoding (MTAD) by combining speculative decoding with multi-token joint decoding (MTJD). MTAD generates high quality draft tokens by estimating the joint distribution of the target model using draft model, while improving both the speed and quality of inference.

**Strengths:**

1. The proposed MTAD guides the generation and verification of draft tokens based on joint probabilities, providing a new idea for speculative decoding.

2. This paper analyzes the speculative decoding technique from an energy perspective for the first time and verifies the energy efficiency of speculative decoding.

3. In addition to the experimental analysis, the authors also theoretically demonstrate the bounded error as well as the effectiveness of the MTAD algorithm.

**Weaknesses:**

1. Despite generating texts with lower perplexity, MTAD is still a lossy speculative decoding method. It is unfair to compare MTAD with lossless speculative decoding methods. It might be useful to add some comparisons with other lossy verification methods (e.g. Medusa's Typical Sampling) to verify the superiority of MTAD.

2. The main experiments in this paper use perplexity as an evaluation criterion for performance, and it is clear that MTAD based on PPL verification will have advantages. More experiments on downstream tasks are needed to verify whether MTAD can really improve the quality of text, since PPL and metrics are not strictly positively correlated.

**Questions:**

1. The threshold for the verification phase of vanilla speculative decoding is sampled from U(0, 1) to ensure losslessness, and MTAD uses a fixed threshold (e.g. 0.1). I'm curious to see if the same speedup can be achieved with the vanilla speculative decoding method using a fixed threshold.

2. The impact from loose thresholds has little effect with PPL and LM-Judger metrics, some more rigorous evaluation (e.g., Humaneval's Pass@K) is needed to validate the methodology.

---

> ### Author Response · Authors · 2024-11-21
> **Response to Reviewer h4X4**
>
> # Comparison with Lossy Verification Methods
> We appreciate your feedback regarding the comparison between MTAD and lossy verification methods. In the original submission we have one lossy baseline, BiLD. Additionally, we have included Medusa's typical sampling algorithm as a baseline. These results, detailed in the "Common Reply on Perplexity and Downstream Performance," demonstrate that MTAD consistently achieves superior efficiency and output quality compared to other lossy verification methods.
>
> # More rigorous evaluation for output quality
> We understand your concern about relying on perplexity (PPL) as a primary performance metric. In our original submission (Table 6), we showed that MTAD delivers improved output quality across multiple datasets using diverse metrics. To address your request for more rigorous evaluations, we have conducted additional experiments on downstream tasks, including HumanEval's Pass@1 and Spider’s execution accuracy (execution accuracy = 1 if the returned SQL result is correct, otherwise it is 0), as shown in the common response. These experiments confirm that MTAD improves not only perplexity but also downstream task quality, bridging the gap between theoretical and practical performance.
>
> # Impact of Fixed Thresholds in Speculative Decoding
> You raised a good question about whether the same speedup achieved by MTAD can be replicated in vanilla speculative decoding using a fixed threshold. To explore this, we evaluated speculative decoding (SpD) with a fixed threshold (thres=0.1), alongside the original lossless SpD and MTAD (thres=0.1). We run Llama-3-8B model (with Llama-3-1B as the draft model) on Spider dataset. The results are summarized below:
>
> |  | *speed* | *#  of tokens* | *EA* |
> | :---- | :---- | :---- | :---- |
> | *Fix-thres SpD* | *37.7* | *3.47* | *16.0* |
> | *SpD* | *31.1* | *2.46* | *21.5* |
> | *MJAD* | *38.9* | *3.78* | *34.0* |
>
> These results demonstrate that while employing a fixed low threshold in vanilla SpD increases acceptance rates and speed, it renders the decoding algorithm lossy, leading to a significant decline in downstream accuracy. Furthermore, even with a fixed threshold, SpD remains less efficient than MTAD. It is because SpD terminates verification immediately after rejecting a token, whereas MTAD continues the process to evaluate whether subsequent tokens can still pass verification.

---

> > ### Author Response · Authors · 2024-11-24
> > **Gentle Reminder: Awaiting Your Response to Our Rebuttal**
> >
> > Dear Reviewer h4X4,
> >
> > I hope this message finds you well. We recently submitted our rebuttal and would like to kindly request your feedback on our responses.
> >
> > We understand that your schedule is demanding and greatly appreciate the time and effort you dedicate to the review process. Your insights are invaluable to us, and we are eager to address any further questions or concerns you may have.
> >
> > Thank you for your attention to this matter. We look forward to your response.
> >
> > Best regards,
> >
> > Authors

---

> ### Comment · Area_Chair_EGgb · 2024-11-25
>
> Dear Reviewer h4X4,
>
> Thank you for your valuable contributions to the review process for the paper! The authors have submitted their rebuttal, and I would greatly appreciate it if you could take a look and provide your response.

---

> ### Comment · Reviewer_h4X4 · 2024-11-27
>
> Thanks for your response. I appreciate your addressing my concerns and will raise my score.

---

> > ### Author Response · Authors · 2024-11-27
> >
> > Thank you so much for your thoughtful feedback and for taking the time to review our additional explanations. We greatly appreciate your recognition of our efforts to address your concerns and your decision to raise your score. Your insights have been invaluable in improving our work, and we sincerely thank you for your support throughout this process.

---

### Official Review · Reviewer_rQ6h · 2024-11-04

**Soundness:** 3
**Presentation:** 2
**Contribution:** 3
**Rating:** 6
**Confidence:** 4

**Summary:**

The paper proposed multi-token assisted decoding by combining multi-token joint decoding and speculative decoding (SpD) to improve the quality of the tokens and inference speed.

**Strengths:**

The paper solves an interesting problem of sampling from a joint distribution which is challenging and proposes an alternate approximate method based on draft model from Speculative Decoding (SpD)

It uses ideas from SpD to also boost the inference speed.

Paper uses an auxiliary model to predict joint distribution of multiple tokens w/ beam sampling to sample the tokens fast.

The power analysis provides some insight into the importance of using SpD

Authors provide a robust set of experiments

**Weaknesses:**

the experiment results can be re-checked, my concern is that lower PPL not always results in better quality, it could also be that the draft model is generating repeated tokens which the target model is accepting, given that there's no guarantee that the draft tokens accepted will follow the target distribution.

the reduction in energy consumption is a by-product of SpD and could have been mentioned in a sub-section rather a full section.
The acceptance rate of draft model should affect both the token-rate and the energy consumption.

No mention of draft model or target model training and if they are different from the baseline algorithms draft and target model in the experiments section.

**Questions:**

target model (larger model) is fine-tuned on chat-GPT data for proposed method, is the same fine-tuned model used for other Speculative Decoding (SpD) methods as well?

What about the draft model? is that fine-tuned as well or used as is?

Can qualitative generations be shown in the paper? There are cases when lower PPL is observed during SpD as the draft and target model are both generating and accepting repeated tokens.

Did the author also check performances (PPL, speed-up) when using a larger draft model?

From table 4., it seems like for past SpD methods there is no correlation between high speed (token/s) will lead to low energy (Joule/s), but that doesn't seem to make sense, as both high speed and low energy should be related to high acceptance rate which leads to small number of target model calls. Any comment/thoughts on this?

Can the authors also add plots on acceptance threshold vs acceptance rate? Given the high token-rates shown in Table 4., I am curious to see how the acceptance rates look like.

---

> ### Author Response · Authors · 2024-11-21
> **Response to Reviewer rQ6h**
>
> # Relationship Between Perplexity and Output Quality, and Qualitative Generations
>
> We appreciate your thoughtful feedback and understand your concerns about the relationship between perplexity and output quality. For a comprehensive explanation, we kindly refer you to our "Common Reply on Perplexity and Downstream Performance," where we provide detailed analyses and additional experiments demonstrating that MTAD's lower perplexity correlates with higher output quality.
>
> We also include qualitative examples below to illustrate that MTAD avoids generating repeated tokens. We provide examples from the HumanEval and Spider datasets using Llama-3-8B and Llama-3-1B models.
>
> **Prompts:**
> ```
> from typing import List
> def filter_by_substring(strings: List[str], substring: str) -> List[str]:
> """ Filter an input list of strings only for ones that contain given substring
> >>> filter\_by\_substring(\[\], 'a')
> []
> >>> filter_by_substring(['abc', 'bacd', 'cde', 'array'], 'a')
> ['abc', 'bacd', 'array']
> """
> ```
> **Output:**
> ```
> return \[str for str in strings if substring in str\]
> ```
> **Prompt:**
> ```
> [INST] <<SYS>> You are a SQL expert. You need to write the correct SQL based on the user question and database schemas. <</SYS>>
> Schema:
> Table concert, columns = [*,concert_ID,concert_Name,Theme,Stadium_ID,Year]
> Table singer, columns = [*,Singer_ID,Name,Country,Song_Name,Song_release_year,Age,Is_male]
> Table singer_in_concert, columns = [*,concert_ID,Singer_ID]
> Table stadium, columns = [*,Stadium_ID,Location,Name,Capacity,Highest,Lowest,Average]
>
> Question: How many singers do we have?
> SQL:
> ```
> **Output:**
> ```
> SELECT count(*) FROM singer;
> ```
> # Use Sub-section for Energy Analysis
> We appreciate your suggestion and we will consider to merge it into a sub-section. Nevertheless, previous studies neglect the energy efficiency of SpD, a critical aspect in the real world. Our analysis addresses this gap and is applicable to MTAD and other SpD approaches, which we believe adds significant value to the community.
>
> # Are Draft and Target Models Different or Fine-tuned?
> We apologize for the lack of clarity on this point. **In all the experiments, our method and all baselines use the same draft and target models.** In all experiments reported in the main paper, we used open-sourced pre-trained draft and target models without any fine-tuning. To further investigate the impact of fine-tuning, we conducted additional experiments (Appendix C.3) where draft models or both draft and target models were fine-tuned.
>
> # Performance with Larger Draft Model
> If you are referring to ablation studies where the target model is fixed while the draft model size varies, we now add an evaluation of  MJAD with Llama-3-8B as the target model and Llama-3-1B and Llama-3-3B as draft models. As expected, Llama-3-3B aligns better with the 8B model, resulting in a higher number of accepted tokens and improved perplexity, which aligns with our analysis in Section 3.2. However, running Llama-3-3B is significantly slower than Llama-3-1B, leading to a noticeable drop in speed.
>
> |  | speed | #  of tokens | ppl |
> | :---- | :---- | :---- | :---- |
> | Llama-3-1B | 33.3 | 4.25 | 1.56 |
> | Llama-3-3B | 18.2 | 4.93 | 1.49 |
>
> # Correlation between Energy and Speed
> We appreciate your question, but we do observe a correlation between speed and energy as shown in the Figure 5, which is newly added to the Appendix C.5, whether considering the entire table or focusing on a specific dataset and model. For fairness,all methods for a given dataset and model were run on the same machine nodes. However, for a fixed method (e.g., Spectr), experiments on different datasets and models might be conducted on different nodes (all equipped with L40 GPUs). We did notice that the same configuration run on different machines may have varied energy consumption. This variation introduces some randomness, which could make the correlation appear less consistent across datasets and models.
>
> In addition, it is possible that a method has a higher acceptance rate but also a higher energy consumption. For example, Spectr increases the acceptance rate by increasing the number of draft sequences, which also increases the cost of verification with the large model. So Spectr might have higher energy consumption in total.
>
> # Acceptance Threshold vs Acceptance Rate
> In Table 5 of the paper, we show the average number of tokens generated per iteration across all datasets. Furthermore, we provide additional results below, showing how the number of tokens per iteration changes with varying acceptance thresholds on the HumanEval dataset using Llama-3 models. These results show that as the acceptance threshold increases, the acceptance rate decreases, leading to fewer tokens generated per iteration.
>
> | thres | 0.1 | 0.3 | 0.5 | 0.7 | 0.9 |
> | :---- | ----: | ----: | ----: | ----: | ----: |
> | Llama-3-8B | 4.72 | 4.43 | 4.12 | 3.93 | 3.64 |
> | Llama-3-8B-Instruct | 4.74 | 4.51 | 4.28 | 4.06 | 3.79 |

---

> > ### Author Response · Authors · 2024-11-24
> > **Gentle Reminder: Awaiting Your Response to Our Rebuttal**
> >
> > Dear Reviewer rQ6h,
> >
> > I hope this message finds you well. We recently submitted our rebuttal and would like to kindly request your feedback on our responses.
> >
> > We understand that your schedule is demanding and greatly appreciate the time and effort you dedicate to the review process. Your insights are invaluable to us, and we are eager to address any further questions or concerns you may have.
> >
> > Thank you for your attention to this matter. We look forward to your response.
> >
> > Best regards,
> > Authors

---

> > > ### Comment · Reviewer_rQ6h · 2024-11-25
> > > **Response to Authors**
> > >
> > > Thanks for the additional experiments and clarification.
> > >
> > > (a) I have a question: in Table 11., do you know why does tokens/sec drop after using fine-tuned 68M draft model? The drop is pretty significant (44.2 -> 25)
> > >
> > > (b) for the acceptance rate vs acceptance threshold table, can the authors also show how the accuracy vary? Is higher acceptance associated with lower accuracy, given that higher acceptance is when the threshold is low which means that the tokens generated are more from draft model than target model?

---

> > > > ### Author Response · Authors · 2024-11-25
> > > >
> > > > Thank you for your questions. We will answer the two questions separately.
> > > >
> > > > (a) We think there are two factors. (1) **Lower Acceptance Rate**: We observed that the average number of accepted tokens decreases by 40%-50% when using the fine-tuned model. This is likely because our fine-tuning process did not explicitly consider the alignment between the small model and the large model. As a result, the fine-tuned draft model may produce outputs that are less compatible with the large model, reducing the efficiency of speculative decoding. (2) **Machine Node Variability**: As mentioned in our rebuttal, all methods for a given dataset and model were run on the same machine node. However, experiments on different datasets and models might have been conducted on different nodes. So for Table 11, different small/large model configurations might be run on different machine nodes. This variation could also contribute to discrepancies in speed.
> > > >
> > > > (b) Thank you for your question. Below, we provide a table that shows how the number of tokens generated per iteration ("acc len") and the output quality (measured as Pass@1) vary with the acceptance threshold. From the table, we can see that a higher threshold leads to a better output quality, which fits the reviewer's intuition. However, as we discussed in Section 5.2.2, when the threshold is too high (say threshold=1), then all the draft tokens are always rejected, then MJAD will be equivalent to multinomial sampling, and the output quality will drop
> > > >
> > > > | Threshold | 0.1  | 0.3  | 0.5  | 0.7  | 0.9  |
> > > > |-----------|-------|-------|-------|-------|-------|
> > > > | **Llama-3-8B acc len** | 4.72  | 4.43  | 4.12  | 3.93  | 3.64  |
> > > > | **Llama-3-8B Pass@1**  | 20.7  | 20.7  | 20.7  | 22.6  | 22.6  |
> > > > | **Llama-3-8B-Instruct acc len** | 4.74  | 4.51  | 4.28  | 4.06  | 3.79  |
> > > > | **Llama-3-8B-Instruct Pass@1**  | 31.7  | 38.4  | 42.1  | 41.5  | 43.3  |

---

> > > > > ### Comment · Reviewer_rQ6h · 2024-11-25
> > > > > **Response to Authors**
> > > > >
> > > > > Thanks a lot for the clarifications, I have updated my ratings

---

> > > > > > ### Author Response · Authors · 2024-11-25
> > > > > >
> > > > > > Thank you so much for your thoughtful suggestions and for taking the time to update your ratings. We deeply appreciate your valuable feedback and support throughout the review process!

---

### Author Response · Authors · 2024-11-21
**Common reply about perplexity and downstream performance**

We acknowledge the reviewers’ concerns regarding whether MTAD’s lower perplexity also translates to improved output quality. In our original submission (Table 6), we demonstrated MTAD’s superior output quality using multiple metrics across diverse datasets with the Llama-2-13B model. **To address these concerns further, we provide additional justification and evaluation below.**

**Multi-Token Joint Decoding (MTJD) has better output quality**
While most decoding approaches focus on inference speed up, we want to design approaches that can also improve inference quality. We propose multi-token joint decoding (MTJD Section 3.1) to achieve the goal, due to its capability to achieve lower perplexity and higher likelihood than single-token multinomial sampling. To validate that MTJD indeed improves output quality, we evaluate MTJD (k=4) and multinomial sampling on Spider, MTBench, and HumanEval (Pass@1) using the Llama-3 series models. We follow the same way introduced in Section 3.1 to implement MTJD. The results, where higher scores indicate better performance, show a clear advantage for MTJD in terms of output quality.

|  |  | Llama-3-8B and Llama-3-1B |  |  | Llama-3-8B-Instruct and Llama-3-1B-Instruct |  |  |
| :---- | :---- | :---- | :---- | :---- | :---- | :---- | :---- |
| dataset |  | Spider | MTBench | HumanEval | Spider | MTBench | HumanEval |
| Single-token multinomial sampling | score | 22.0 | 3.40 | 15.9 | 36.0 | 4.11 | 28.0 |
|  | ppl | 2.58 | 2.40 | 2.09 | 2.23 | 1.91 | 1.85 |
| Multi-token joint sampling | score | 52.5 | 3.77 | 36.6 | 60.5 | 4.40 | 49.4 |
|  | ppl | 1.16 | 1.32 | 1.26 | 1.18 | 1.27 | 1.15 |

Since MTJD is computationally expensive (slower than greedy decoding), we further propose MTAD to approximate MTJD, inheriting its benefits while achieving significantly better efficiency. The performance of MTAD is discussed below.

**Comprehensive Evaluation of MTAD**
To further validate MTAD’s effectiveness, we conducted additional evaluations, addressing reviewers’ requests: (1) we use Llama-3 models (Llama-3.1-8B and Llama-3.1-8B-Instruct as target models, and Llama-3.2-1B and Llama-3.2-1B-Instruct as draft models); (2) we include three datasets: Spider, HumanEval, and MT-Bench, to include rigorous metrics and a variety of tasks; (3) we add Medusa’s \[1\] decoding algorithm, typical sampling, as an additional baseline. **The results below show that MTAD consistently achieves the best output quality and the fastest speed across all datasets and models.**

|  |  |  | Lossy Speculative Decoding |  | Lossless Speculative Decoding |  |  | | Ours |
| :---- | :---- | :---- | :---- | :---- | :---- | :---- | :---- | :---- |:----|
|  |  |  | BiLD | Typical (Medusa) | spec | spectr | specinfer|MCSS | MTAD |
| Spider | Llama-3-Instruct | tokens/s | 20.1 | 22.3 | 19.6 | 22.4 | 21.1 | 21.7| **23.5** |
|  |  | J/token | 10.2 | 9.5 | 10.5 | 9.6 | 10.2 |10.0 |**9.2**|
|  |  | Acc | 35.0 | 42.0 | 36.0 | 35.5 | 37.0 |35.0|**44.0**|
|  | Llama-3 | tokens/s | 23.3 | 32.3 | 31.1 | 32.1 | 32.6 |32.7| **33.3**|
|  |  | J/token | 8.2 | 7.9 | 7.5 | **7.1** | 8.1 | 8.0|7.8 |
|  |  | Acc | 30.5 | 29.5 | 21.5 | 23.0 | 21.5 |24.0| **35.0** |
| MT-Behcn | Llama-3-Instruct | tokens/s | 25.9 | 23.4 | 26.0 | 26.2 | 26.3 |26.8| **29.8** |
|  |  | J/token | 10.8 | 12.2 | 10.0 | 9.9 | 10.0 |9.9| **9.2** |
|  |  | score | 4.15 | 4.26 | 4.1 | 4.11 | 4.01 |4.02| **4.40** |
|  | Llama-3 | tokens/s | 24.5 | 22.3 | 24.1 | 24.5 | 24.5 |25.7| **28.2** |
|  |  | J/token | 11.5 | 12.4 | 11.0 | 11.6 | 11.7 | 11.1|**10.0** |
|  |  | score | 3.41 | 3.24 | 3.39 | 3.41 | 3.35 |3.36| **3.75** |
| HumanEval | Llama-3-Instruct | tokens/s | 17.4 | 21.7 | 22.2 | 23.8 | 22.8 |23.7| **24.8** |
|  |  | J/token | 10.0 | 8.1 | 7.8 | 7.8 | 7.9 |7.8| **7.6** |
|  |  | pass@1 | 37.8 | 35.9 | 32.9 | 32.9 | 31.0 |32.0|**38.4**|
|  | Llama-3 | tokens/s | 19.6 | 22.5 | 22.2 | 24.4 | 22.5 |23.8|**25.6**|
|  |  | J/token | 9.7 | 8.9 | 8.9 | 8.9 | 8.1 |7.9|**7.6**|
|  |  | pass@1 | 19.5 | 20.0 | 15.9 | 16.0 | 17.7 |17.0| **22.0** |

**Visualization of relationship between perplexity and downstream performance**

To further illustrate the relationship between perplexity and downstream performance, we present a scatter plot (Figure 4 in the Appendix C.4). The plot shows the correlation between **relative downstream scores** (normalized by the score of multinomial sampling) and **relative perplexity** (normalized by the perplexity of multinomial sampling) across 7 decoding algorithms, 3 datasets, and 2 model configurations. The results confirm that lower perplexity generally correlates with higher output quality.

We sincerely hope these additional results and analyses could address the reviewers’ concerns and demonstrate the robustness, efficiency, and quality improvements achieved by MTAD.

\[1\] Cai, Tianle, et al. "Medusa: Simple llm inference acceleration framework with multiple decoding heads." *arXiv preprint arXiv:2401.10774* (2024).

---

> ### Author Response · Authors · 2024-11-22
>
> Dear Reviewers,
>
> We hope this message finds you well. We have submitted a comprehensive evaluation demonstrating that our method enhances output quality while being the fastest and most energy-efficient. Additionally, we have addressed all specific questions raised.
>
> If you have any further inquiries or require additional information, please let us know. We are committed to providing any necessary clarifications before the discussion phase concludes.
>
> Thank you for your time and consideration.
>
> Best regards,
> Authors

---

> ### Comment · Reviewer_rQ6h · 2024-11-25
> **Question for Authors**
>
> Thanks for the additional results.
>
> Do you know why do the lossless methods in above table have different accuracies? From my understanding the lossless speculative methods are known to maintain the accuracy with the accuracy of autoregressive target model.
> Even if we run the target model in sample-mode, if seed is fixed, then the outputs generated should be same, which should result in same accuracy.

---

> > ### Author Response · Authors · 2024-11-25
> > **Why lossless method have different accuracy**
> >
> > Thank you for your question. You raise an important point, but there seems to be a slight misunderstanding regarding the behavior of lossless speculative methods. Specifically, it is not necessarily true that "if the seed is fixed, the outputs generated by lossless methods should be the same." It is only true if the target distribution is warped by argmax (i.e., deterministic greedy decoding).
> >
> > Let us consider the case where only one token is to be generated, with the target distribution denoted as $p$ and the draft distribution as $q$. **While lossless speculative methods ensure that the output tokens follow the target distribution $p$, their internal mechanisms introduce variability that can lead to different outputs even with the same seed.**
> >
> > For example:
> >
> > 1. **Multinomial Sampling**: The output token is directly sampled from $p$. Since this involves a single sampling step, fixing the random seed ensures consistent outputs.
> >
> > 2. **Vanilla Speculative Decoding**: This method first samples a draft token $x$ from $q$ and an acceptance threshold $r$ from $U(0, 1)$. If $x$ is accepted ($p(x)/q(x) > r$), the output token is $x$. If rejected ($p(x)/q(x) \leq r$), the output token is sampled from the residual distribution $ \text{norm}(\max(p - q, 0)) $. This introduces additional variability beyond the fixed seed.
> >
> > 3. **SpecInfer**: This method samples multiple draft tokens (e.g., two) for the same position from $q$. The first draft token is accepted or rejected similarly to vanilla speculative decoding. If rejected, the second draft token undergoes a slightly different acceptance check. If both are rejected, the output token is sampled from a different residual distribution. Each step introduces variability, even with a fixed seed.
> >
> > While all these methods ensure that the output tokens follow the target distribution $p$, it is evident that fixing the same random seed cannot guarantee identical outputs across methods due to differences in their internal processes.
> >
> > As a result, it is normal for the accuracies of lossless methods to show slight variations. However, as observed in the results, these differences are generally minimal and remain within a close range, reflecting their shared objective of adhering to the target distribution.

---

> ### Comment · Reviewer_v4L1 · 2024-11-25
> **Additional Questions for Authors**
>
> Thanks for the additional results. Here I have further questions about the results.
>
> What is the sampling method you used in Specinfer method to construct the token trees, e.g., with replacement sampling or without replacement sampling?

---

> > ### Author Response · Authors · 2024-11-25
> > **SpecInfer details**
> >
> > Thank you for your question. We follow the SpecInfer paper and use sampling **with replacement** to construct the draft token trees. Notably, a recent preprint study [1] suggests that implementing sampling **without replacement** to form draft token tree while maintaining losslessness requires non-trivial modification.
> >
> > [1] Jeon, Wonseok, et al. "Recursive speculative decoding: Accelerating LLM inference via sampling without replacement." *arXiv preprint* arXiv:2402.14160 (2024).

---

> > > ### Comment · Reviewer_v4L1 · 2024-11-25
> > > **Response to Authors**
> > >
> > > Thank you very much for your response.
> > >
> > > > Notably, a recent preprint study [1] suggests that implementing sampling without replacement to form draft token tree while maintaining losslessness requires non-trivial modification.
> > >
> > > From my perspective, it can be easily implemented with just a few code modifications, so describing it as "non-trivial" seems inaccurate.
> > >
> > > Moreover, using sampling without replacement to construct draft token trees generally yields better results compared to the with-replacement method in most cases. Notably, state-of-the-art methods like Eagle [1], which leverage the SpecInfer approach, also adopt the without-replacement sampling method for token tree construction.
> > >
> > > I will appreciate it, if you could consider conducting further experiments with sampling without replacement.  This would provide a more accurate comparison to demonstrate how your methods outperform existing state-of-the-art approaches.
> > >
> > > [1] EAGLE: Speculative Sampling Requires Rethinking Feature Uncertainty.

---

> ### Author Response · Authors · 2024-11-25
> **Additional comparision with Spectr and SpecInfer (sample without replacement)**
>
> Thank you for your thoughtful suggestions. We appreciate your perspective and conduct experiments incorporating sampling without replacement as suggested. The detailed results are included in our separate response to you (reviewer v4L1). The results show that the effect of sampling without replacement is not significant, and MJAD is still the fastest method with highest output quality.
>
> However, we want to clarify some issues regarding to sampling without replacement:
>
> In Spectr and SpecInfer, both methods assume that draft tokens are sampled **independently** to ensure the output token aligns with the target distribution (Theorem 4.2 in the SpecInfer paper and Theorem 2 in Spectr paper). **Sampling without replacement breaks this assumption because the probability of selecting a token is affected by previous selections.** While this deviation may not significantly impact output quality in practice, it does invalidate the assumption that each draft sample is drawn independently. As a result, the formal losslessness guarantee cannot be directly extended to the without-replacement case without revisiting and modifying the theoretical framework.
>
> We hope this explanation and additional results address your concerns, and we sincerely appreciate your thoughtful feedback and encouragement.

---

> > ### Comment · Reviewer_rQ6h · 2024-12-02
> > **Reply to authors**
> >
> > Just to add on to above, I think Recursive Speculative Decoding [1] shows theoretically how to sample from draft tree without replacement and could be useful for the authors in future.
> >
> > [1] Recursive Speculative Decoding: Accelerating LLM Inference via Sampling Without Replacement

---

> > > ### Author Response · Authors · 2024-12-02
> > >
> > > Thank you for the thoughtful suggestions. We will gladly include [1] in our related work. We also appreciate the reviewer bringing the MCSS paper to our attention. Upon reviewing it, we found that MCSS and recursive sampling decoding share many similarities, and incorporating this additional baseline has made our experiments more robust. Furthermore, we agree that extending our method to support multiple draft sequences and sampling without replacement presents an exciting opportunity for future improvement, and we will highlight this as a potential direction for further exploration.

---

### Meta-Review · Area_Chair_EGgb · 2024-12-26

**Metareview:**

The paper introduces Multi-Token Assisted Decoding (MTAD), a speculative decoding framework combining multi-token joint decoding (MTJD) with speculative decoding (SpD).  The proposed framework demonstrates significant gains in perplexity (21.2% reduction), task performance, speed-up (1.42×), and energy efficiency (1.54×) across multiple datasets.

**Strengths** (1) The paper improves speculative decoding with multi-token joint decoding, which advances beyond single-token-based methods. A theoretical analysis of bounded error provides additional insight into the method. (2) Comprehensive evaluation of multiple datasets and tasks.

**Weaknesses** (1) Limited comparison with recent baselines and qualitative analysis (2) Some reviewers raised concerns about the relationship between perplexity and quality

**Decision** Considering the novelty of the proposed methods, and how all the reviewers found the rebuttal addressing their original concerns, I am leaning toward acceptance.

**Additional Comments On Reviewer Discussion:**

The author provided more experimental results with further explanations in the rebuttal phase. All the reviewers are satisfied with the updated results and uniformly adjusted their scores to positive to indicate the acceptance.

---

### Decision · Program_Chairs · 2025-01-22

Accept (Poster)